

# Seasonal evolution and parameterization of Arctic sea ice bulk density: results from the MOSAiC expedition and ICESat-2/ATLAS

Yi Zhou[1,2], Xianwei Wang[1,2], Ruibo Lei[3], Luisa von Albedyll[4], Donald K. Perovich[5], Yu Zhang[6], Christian Haas[4]

[1]School of Oceanography, Shanghai Jiao Tong University, Shanghai, China
[2]Key Laboratory of Polar Ecosystem and Climate Change (Shanghai Jiao Tong University), Ministry of Education, Shanghai, China
[3]Key Laboratory for Polar Science, Ministry of Natural Resources, Polar Research Institute of China, Shanghai, China
[4]Alfred Wegener Institute, Helmholtz Centre for Polar and Marine Research, Bremerhaven, Germany
[5]Thayer School of Engineering, Dartmouth College, Hanover, NH, USA
[6]College of Oceanography and Ecological Science, Shanghai Ocean University, Shanghai, China

**Correspondence:** Xianwei Wang (xianwei.wang@sjtu.edu.cn) and Ruibo Lei (leiruibo@pric.org.cn)

**Abstract.** Satellite retrievals of Arctic sea ice thickness typically assume a constant sea ice bulk density (IBD), overlooking its seasonal variations influenced by ice internal texture and contaminants. This study unveils the initial insights into the seasonal evolution and parameterization of IBD during the Arctic freezing season from October to April. To retrieve IBD, we combined in situ observations obtained from ice mass balance buoys, snow pits, and snow transects during the Multidisciplinary drifting Observatory for the Study of Arctic Climate (MOSAiC) expedition, as well as laser freeboard data derived from the Ice, Cloud, and Land Elevation Satellite-2 (ICESat-2). Assuming hydrostatic equilibrium, local-scale IBDs for the level ice component of the MOSAiC ice floes, predominantly consisting of second-year ice, were obtained at a spatial scale of approximately 50 km. The results indicated a statistically significant seasonal decreasing trend in IBD at a rate of ~16 kg m$^{-3}$ per month ($P < 0.001$) from mid-October to mid-January, likely attributable to increased internal porosity as the sea ice aged. This was followed by a relatively stable period from mid-January to mid-April, with an average IBD of ~897 ± 11 kg m$^{-3}$. Core-based IBDs from eight MOSAiC sites showed a similar seasonal pattern, but with a narrower range of variation and an earlier onset of the relatively stable period, possibly owing to the spatial heterogeneity of the MOSAiC ice floes. Based on regression analyses, we developed updated parameterizations for IBD that are anticipated to be applicable throughout the freezing season, encompassing both first- and second-year ice. In particular, the ice draft-to-thickness ratio emerged as the most efficient parameter for determining IBD ($R^2 = 0.99$, $RMSE = 1.62$ kg m$^{-3}$), with potential application to multi-year ice and deformed ice as well. Our updated parameterizations have the potential to optimize basin-scale satellite-derived sea ice thickness, thereby contributing to more accurate monitoring of changes in sea ice volume.



## 1 Introduction

Sea ice thickness is a crucial variable for Arctic sea ice, indicating its overall physical state in the context of a warming climate (Sumata et al., 2023). Various techniques have been utilized to measure ice thickness, such as sea ice mass balance buoys (IMBs, Perovich et al., 2014), helicopter- or ship-based electromagnetic induction (Haas, 1998; Haas et al., 2009), upward-looking sonars (Rothrock et al., 2008), anchored moorings (Hansen et al., 2013), and satellite altimeters (Landy et al., 2022). In the present era, satellite altimeters have become the primary means of determining ice thickness in the Arctic

Ocean thanks to their ability to provide basin-scale and long-term observations. Assuming hydrostatic equilibrium, ice thickness can be converted from the freeboard data obtained from satellite altimeters, which necessitates prior knowledge of sea ice bulk density (IBD) and snow loading (Laxon et al., 2013). Nevertheless, it should be noted that satellite-based ice thicknesses may contain notable uncertainties due to a limited understanding of the spatial and temporal variations of IBD and snow mass (Zygmuntowska et al., 2014; Kern and Spreen, 2015). The estimated ice thicknesses derived from CryoSat-2

altimetry data exhibit systematic uncertainties of up to 0.6 m in first-year ice (FYI) and 1.2 m in multi-year ice (MYI) (Ricker et al., 2014). These significant uncertainties impede the validation of model predictions and the assessment of the changing Arctic climate system (Gerland et al., 2019; Notz and Community, 2020).

Compared to snow mass, uncertainty in IBD may have a more significant impact on the total ice thickness error (Zygmuntowska et al., 2014; Kern et al., 2015; Kwok and Cunningham, 2015; Landy et al., 2020). As a result, it is regarded

as a pivotal factor in accurately converting ice freeboard to thickness. Specifically, IBD accounts for 30–35% of the average absolute uncertainty in ICESat-based ice thickness, with higher uncertainty in autumn than in spring due to lower snow accumulation (Zygmuntowska et al., 2014). For radar altimeter-based ice thickness estimates, variations in IBD settings can lead to thickness deviations of more than 0.5 m (Kern et al., 2015). Consequently, accurate ice thickness measurements from space necessitate IBD retrievals with adequate spatial representativeness and seasonal variability. The substantial

heterogeneity in the internal structure of sea ice and the significant seasonal variation in its porosity, linked to ice growth and decay processes, underscore the importance of this aspect (Perovich et al., 2001; Petrich and Eicken, 2010; Oggier and Eicken, 2022).

Sea ice has a complex and heterogeneous structure, comprising multiphase substances such as ice crystals, liquid brine, and air bubbles (Pustogvar and Kulyakhtin, 2016). Conventional techniques for measuring IBD typically involve

mass/volume, submersion, and specific gravity methods, which require sampling, ice block preparation, and measurement (Timco and Frederking, 1996). During the sampling process, the sample size and brine drainage process can introduce significant errors in the calculation of IBD (Pustogvar and Kulyakhtin, 2016). To reduce measurement errors and sampling constraints, several studies have used the airborne multi-sensor data to estimate IBD based on the hydrostatic equilibrium method (Alexandrov et al., 2010; Jutila et al., 2022). This method has effectively prevented sampling errors and captured

IBD variability across different ice types within the airborne observation footprint (Pustogvar and Kulyakhtin, 2016). However, due to logistical challenges, airborne observations are predominantly limited to late spring. As sea ice is strongly



modulated by its internal structure and porosity, attributable to desalination, snow-ice formation or deformed ice, as well as changes in the thickness ratio between residual and newly formed ice layers (Golden et al., 2007; Ishii and Toyota, 2012; Lei et al., 2022; Oggier and Eicken, 2022), these factors collectively contribute to the seasonal variability of IBD. Therefore, the accurate representation of this seasonal evolution necessitates the development of more reliable parameterization methods, which can be achieved through the acquisition of continuous observational data.

Given the significant knowledge gap in IBD, ice thickness retrievals have primarily relied on the IBD climatology derived from limited field measurements (Alexandrov et al., 2010; Quartly et al., 2019; Ji et al., 2021). For most CryoSat-2 ice thickness products, the IBD climatology developed by Alexandrov et al. (2010) (referred to as A10) serves as the main input for the freeboard-to-thickness conversion (Sallila et al., 2019). The FYI density of A10 was determined by the hydrostatic equilibrium method with an estimated value of $917 \pm 36$ kg m$^{-3}$. By contrast, given the significant disparity in ice density and porosity between exposed and submerged MYI, the MYI density of A10 ($882 \pm 23$ kg m$^{-3}$) was calculated by considering the ice portions above and below the sea surface and weighted by the corresponding reference IBDs (Timco and Frederking, 1996; Pustogvar and Kulyakhtin, 2016). However, Shi et al. (2023) have recently argued that the reference IBD of MYI above the sea surface, defined in A10 as only 550 kg m$^{-3}$, does not accurately represent the properties of MYI, highlighting the potential uncertainties of the A10 approach. Furthermore, several studies have examined the relationship between IBD and other sea ice parameters, with the goal of developing IBD parameterizations. For instance, a quadratic correlation between IBD and sea ice thickness (Kovacs, 1997) and an exponential relationship with sea ice freeboard (Jutila et al., 2022) have been identified. However, the applicability and robustness of these parameterizations in estimating IBD are uncertain, as they were derived from limited field data collected mainly during the late freezing season.

The recent synergistic observations from the Multidisciplinary drifting Observatory for the Study of Arctic Climate (MOSAiC) expedition and NASA's Ice, Cloud, and Land Elevation Satellite-2 (ICESat-2, IS2) provide a unique opportunity to estimate IBD throughout the Arctic freezing season. During the MOSAiC expedition, a variety of IMBs capable of simultaneously measuring both snow depth and ice thickness were deployed (Lei et al., 2022; Perovich et al., 2023). Furthermore, field measurements of snow transects (Itkin et al., 2023), snow pits (Macfarlane et al., 2023), and ice cores (Angelopoulos et al., 2022) provided additional information on snow depth, snow bulk density, and core-based IBD data. From space, IS2 carried the Advanced Topographic Laser Altimeter System (ATLAS), which provides track-based height measurements of snow and sea ice (hereafter total freeboard), with an uncertainty of approximately 2–4 cm (Kwok et al., 2019a; Kwok et al., 2019b). With the appropriate resampling of the IS2 track-based freeboards to obtain local-scale freeboard data, it would theoretically be possible to estimate the IBD of the MOSAiC ice floes using the hydrostatic equilibrium method by combining the IMBs and other MOSAiC field measurements.

In this study, we integrated IS2 along-track freeboards, IMB data, and in situ observations from snow transects and snow pits at different spatial scales to retrieve IBD (Section 3.1). We aim to investigate the seasonal evolution of IBD during the freezing season (Section 3.2), a period considered the most reliable for retrieving ice thickness (Ricker et al., 2014; Petty et al., 2020), and to develop updated parameterizations for IBD to improve its seasonal and spatial representativeness (Section



3.3). Furthermore, we also discussed the uncertainties and limitations in the IBD retrieval process (Section 4.1), explored potential factors influencing the seasonal evolution of IBD (Section 4.2), revealed the spatial heterogeneity of IBD (Section 4.3), assessed the potentially broader applicability of our updated IBD parameterizations (Section 4.4), and highlighted the prospects for pan-Arctic IBD estimates (Section 4.5).

## 2 Data and Methods

### 2.1 Data

We used snow depth and ice thickness data from 15 IMBs, snow bulk density data from snow pits, and along-track freeboard data from IS2/ATLAS to retrieve IBD. Additionally, core-based IBD data and snow depth data from snow transects were included in the comparative analysis. All data were collected between October 2019 and April 2020 during the MOSAiC expedition.

#### 2.1.1 MOSAiC observations

The MOSAiC expedition was conducted in 2019−2020 and focused on studying the atmosphere, sea ice, ocean, and ecosystems in the Arctic Ocean (Nicolaus et al., 2022; Rabe et al., 2022; Shupe et al., 2022). The research vessel *Polarstern* was anchored on an ice floe, facilitating a comprehensive year-long Lagrangian study along the Transpolar Drift (see Fig. 1a). A distributed network (DN) of autonomous IMBs was deployed within 30–40 km of the ship and the central observatory (CO). The measurements were mainly performed on the residual ice that survived the summer of 2019 and transformed into second-year ice (SYI) by autumn 2020 (Krumpen et al., 2020; Krumpen et al., 2021).

**IMB data.** Within the MOSAiC DN, the Snow and Ice Mass Balance Array (SIMBA, abbreviated as T) and the Seasonal Ice Mass Balance Buoy (SIMB, abbreviated as I) are the principal instruments for automated measurements of snow and sea ice mass balance (Lei et al., 2022; Perovich et al., 2023). The SIMBAs and SIMBs were deployed over the level ice, devoid of melt ponds, to ensure the reliability and representativeness of the buoy data (see details in Table S1). The SIMBAs are able to record the vertical temperature profile within the snow/ice system and detect thermal changes in the vicinity of thermistors following pulse heating events. The integration of these measurements allows the determination of ice thickness and snow depth at SIMBA sites (Provost et al., 2017; Liao et al., 2018). The SIMBs incorporate a thermistor string similar to that of the SIMBAs, as well as ranging sonar and meteorological sensors (Polashenski et al., 2011). In this study, the daily resampled snow depth and ice thickness data from 12 SIMBAs and 3 SIMBs were used to retrieve IBDs (Fig. 1b). We selected these buoys from a more extensive set because they have a prolonged operational duration and a higher spatial overlap with the IS2 data. These measurements comprise 4 buoys deployed in the FYI and 11 in the SYI. In particular, the selected T72 buoy recorded the formation of snow-ice (see Fig. 4a).



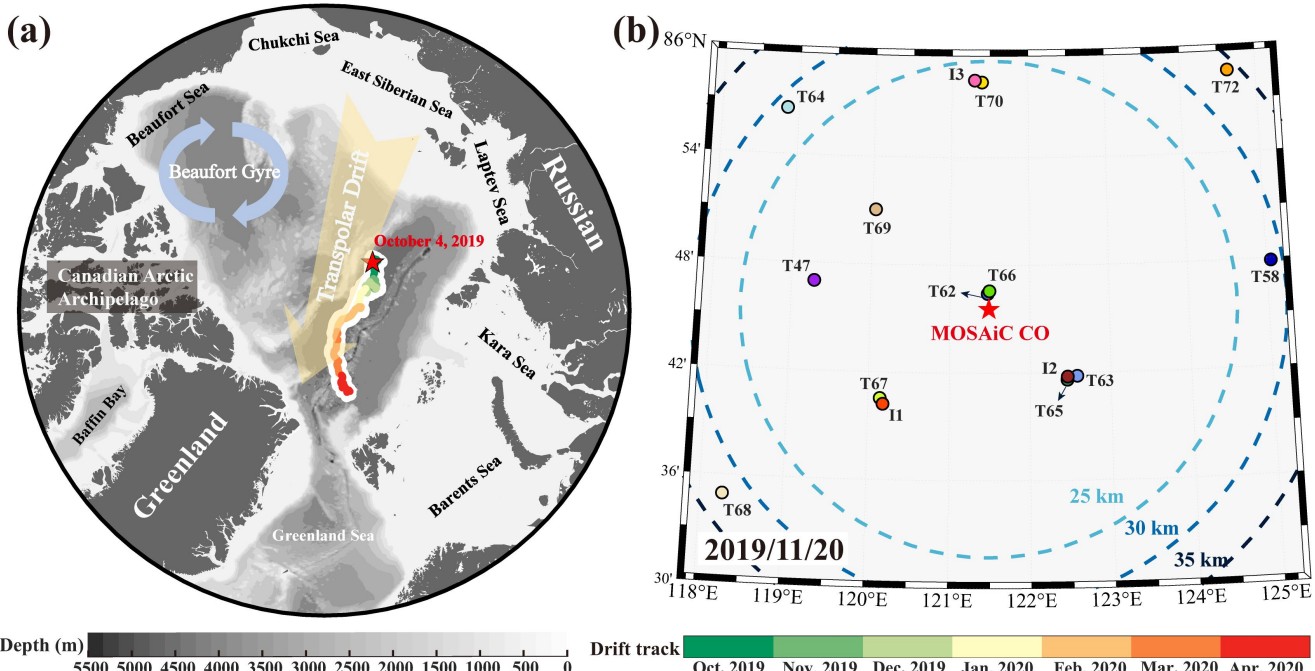

**Figure 1. (a)** Drift trajectory of the MOSAiC CO, with the starting site marked by a red pentagram (4 October, 2019). The background field shows the bathymetry data from ETOPO1. **(b)** Locations of the 12 SIMBAs and 3 SIMBs in the vicinity of the MOSAiC CO on 20 November 2019.


**Snow transect and snow pit data.** Snow depth and bulk density measurements are crucial for calculating IBD based on the principle of hydrostatic equilibrium. During the MOSAiC expedition, field snow depth data were systematically collected along transects at intervals of approximately once or twice per week throughout the year using a *Magnaprobe* (Sturm and Holmgren, 2018), yielding an extensive snow depth dataset over the CO floe (Itkin et al., 2023). Snow measurements were

spaced horizontally between 1 and 3 m and varied with surface roughness and ice type along the transect lines. Considering the spatial heterogeneity of the snow cover, daily snow depth data were compiled from the level ice components across all repeated transect lines and loops, and then compared with buoy-measured mean snow depths. Furthermore, extensive field snow pit measurements (576 in total) were also conducted at the CO floe (Macfarlane et al., 2023). The snow density was determined using the traditional gravity-volume method. The average density of all layers across all available snow pits on a

given day was calculated to represent the snow bulk density.

**Ice core data.** Ice cores for FYI and SYI were collected from eight sites during MOSAiC, including two year-round coring sites, three temporary coring sites on the CO floe, and the DN L1, L2, and L3 sites. The sampling sites provided detailed insights into seasonal variations in ice temperature, salinity and density (Angelopoulos et al., 2022). In contrast to the other

sampling sites, which commenced data collection in late October, the DN sites collected cores primarily in early October,



thereby enhancing the temporal availability of ice core data. The sea ice density was determined from the core samples using the hydrostatic weighing method at a laboratory temperature of −15 °C. This study calculated the average density of all ice layers to represent the core-based IBD, which was then compared to the local-scale IBD over the MOSAiC ice floes.

### 2.1.2 IS2 along-track freeboard data

The IS2 satellite is equipped with the ATLAS instrument, which employs a low-pulse-energy laser configured into six beams that are split into three pairs of varying intensities for detailed surface mapping (Kwok et al., 2019a; Kwok et al., 2019b). The strong beams possess about four times the pulse energy of the weak beams, thereby achieving a higher spatial resolution. The vertical accuracy of IS2 elevation measurements ranges from approximately 7 to 10 cm, with a ground footprint of about 17 m (Markus et al., 2017; Kwok et al., 2019a). This study used the ATL10 freeboard dataset (L3A, version 5) from the

National Snow and Ice Data Center (NSIDC) to retrieve the local-scale IBD, which offers along-track freeboard measurements from both strong and weak beams. For each ground track, we exclusively utilized the data from the strong beams due to their enhanced along-track resolution in segment length (Shen et al., 2020; Petty et al., 2023). A 150-segment weighted average was then applied to these freeboard measurements to reduce data noise and computational burden, according to Petty et al. (2020).

## 2.2 Methods

This section presents the methods used to resample the IS2 along-track freeboard (Section 2.2.1), the retrieval of the local-scale IBD (Section 2.2.2), the uncertainty in the calculation of the IBD (Section 2.2.3), and the principles for developing the IBD parameterization (Section 2.2.4). Figure 2 depicts the detailed steps involved in retrieving IBD, and these processes are further mentioned in the sections that follow.

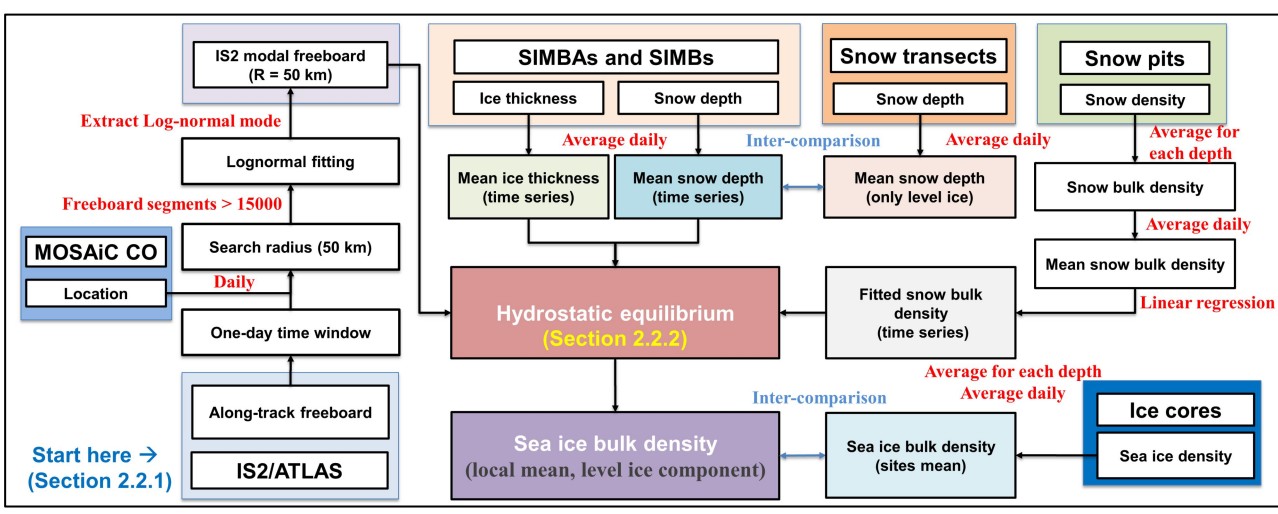


**Figure 2.** Flowchart of the IBD retrieval process.



### 2.2.1 IS2 modal freeboard

In order to estimate the IBD, it is essential to obtain a local-scale freeboard representative of the MOSAiC DN ice floes. This was achieved in this study by a simple modal approach (see left column of Fig. 2): (1) Valid IS2 along-track freeboards were identified within a specific search radius centered on the MOSAiC CO. (2) A log-normal fit was applied to the identified IS2 freeboard distribution. (3) The mode of the fitted log-normal distribution was determined as the representative freeboard of the level ice, termed the "IS2 modal freeboard". An illustrative example of resampling the IS2 along-track freeboards to obtain the local-scale freeboard data is shown in Fig. 3.

Two principal arguments support the feasibility of this method. Firstly, at the local scale, the freeboard of Arctic sea ice conforms to a log-normal (or exponential) distribution (e.g., Haas, 2010; Farrell et al., 2011; Petty et al., 2016; Landy et al., 2019). Secondly, the modal freeboard derived from satellite along-track measurements well captures the ice growth driven by thermodynamic forcing (Ricker et al., 2015; Koo et al., 2021). Fundamentally, the IS2 modal freeboard is a reliable indicator of level ice thickness, excluding the impacts of deformed ice components. This property is advantageous for estimating the IBD of level ice using the hydrostatic equilibrium method (Hutchings et al., 2015). The details of the resampling process for the IS2 along-track freeboards are as follows (see also Fig. 2):

First, we perform a nonlinear least-squares fit of a log-normal function to the IS2 along-track freeboards within a specified search radius centered on the MOSAiC CO:

$$\varphi\left(h_{\mathrm{f}}^{\mathrm{R}}\right) = \frac{1}{h_{\mathrm{f}}^{\mathrm{R}} \sigma \sqrt{2\pi}} \times \exp\left(\frac{-(\ln{(h_{\mathrm{f}}^{\mathrm{R}})} - \mu)^2}{2\sigma^2}\right), \tag{1}$$

where $\varphi\left(h_{\mathrm{f}}^{\mathrm{R}}\right)$ represents the estimate of the log-normal fit, $h_{\mathrm{f}}^{\mathrm{R}}$ denotes the IS2 along-track freeboard within the specified search radius, and $\mu$ and $\sigma$ are the mean and standard deviation of the log-normal function, respectively. In this study, we set the search radius to 50 km to identify valid IS2 along-track freeboards, as this scale fully encompasses the measurement range of the MOSAiC DN and facilitates sufficient IS2 along-track freeboard segments to fit the log-normal distribution.

Second, the IS2 modal freeboard ($\mathrm{Mode}\,[h_{\mathrm{f}}^{\mathrm{R}}]$) is derived from the fitted log-normal distribution:

$$\mathrm{Mode}\,[h_{\mathrm{f}}^{\mathrm{R}}] = \exp{(\mu - \sigma^2)}, \tag{2}$$

Here, the IS2 modal freeboard ($\mathrm{Mode}\,[h_{\mathrm{f}}^{\mathrm{R}}]$) was only calculated if the valid IS2 along-track freeboard segments exceeded 15000 (Fig. 2).

We acknowledge that there are inevitable spatio-temporal discrepancies in matching IS2 along-track freeboards to the MOSAiC DN measurements, attributable to ice advection and its heterogeneous nature (Dethloff et al., 2022). Nevertheless, it remains confident that the MOSAiC DN local-scale of ~30−40 km offers sufficient spatial representation for matching with satellite measurements. Furthermore, the log-normal mode, which closely aligns with the IS2 dominant freeboard (i.e., the frequency maxima), effectively represents the level ice freeboard at this scale (Fig. 3). However, due to the potential for the multi-modal distribution of freeboards creating a quasi-peak region (Fig. 3), the dominant freeboard was not selected as



the representative freeboard. This situation suggests that the frequency distribution (FD) resolution settings can significantly influence the ultimate frequency maxima. Overall, we conclude that employing the mode of the log-normal distribution to

identify the representative freeboard provides better reliability than utilizing the dominant freeboard. Based on this, a preliminary uncertainty in the IS2 modal freeboard ($\sigma_{\mathrm{mode}}$) was delineated, characterized by the absolute difference between the log-normal mode and the dominant freeboard in the FD (Fig. 3):

$$\sigma_{\mathrm{mode}} = \left| \mathrm{Mode}\left[ h_f^R \right] - f_{\max}\left[ h_f^R \right] \right|. \tag{3}$$

where the $f_{\max}\left[ h_f^R \right]$ represents the dominant freeboard in the FD, determined using a resolution (or bin size of the FD) of

0.01 m.

### 2.2.2 Retrieval of IBD

By integrating the ice thickness, snow depth and bulk density obtained during the MOSAiC expedition and the IS2 modal freeboard, we adopt the hydrostatic equilibrium method to retrieve the IBD (e.g., Alexandrov et al., 2010; Jutila et al., 2022), as shown in Fig. 2:

$$\rho_i h_i + \rho_s h_s = \rho_w (h_i - h_f + h_s), \tag{4}$$

where $\rho_i$ is the IBD, $\rho_w$ and $\rho_s$ represent the seawater and snow bulk densities, respectively; $h_i$, $h_f$, and $h_s$ are the sea ice thickness, total freeboard, and snow depth, respectively. Thus, the IBD can be estimated as:

$$\rho_i = \rho_w \left( 1 - \frac{h_f}{h_i} + \frac{h_s}{h_i} \right) - \rho_s \frac{h_s}{h_i}. \tag{5}$$

In this study, $h_i$ and $h_s$ were obtained from the SIMBAs and SIMBs; $h_f$ was denoted as the IS2 modal freeboard

( $\mathrm{Mode}\left[ h_f^R \right]$ ); $\rho_w$ was set to 1024 kg m$^{-3}$ following Wadhams et al. (1992), and $\rho_s$ was obtained from the snow pit measurements (Macfarlane et al., 2023), see details in Fig. 2. This method provides an estimate of the level ice mixed with FYI (a small fraction) and SYI (a large fraction) at the MOSAiC local-scale (~50 km). To the best of our knowledge, these are the first estimates of seasonal variability throughout the freezing season in the IBD of relatively young sea ice less than 16 months old (Krumpen et al., 2020). In the following analysis, the local-scale IBD of the MOSAiC ice floes, derived

mainly from the SIMBAs (or SIMBs) and IS2/ATLAS, are referred to as "SI".



**Figure 3.** Illustrative example of resampling the IS2 along-track freeboards within a 50-km radius of the MOSAiC CO, showing the IS2 along-track freeboard (150 segment mean) and the corresponding freeboard frequency distribution (FD) on **(a)** 11 October 2019, **(b)** 9 December 2019, and **(c)** 31 January 2020. Each FD panel shows the valid IS2 freeboard segments (Num), the IS2 modal freeboard and its uncertainty (Unc.).





### 2.2.3 Uncertainty in IBD retrieval

The uncertainty in the retrieved IBD ($\sigma_{\rho_i}$) is determined using the Gaussian error propagation method, assuming that the uncertainties of the individual variables in Eq. (5) are independent (Jutila et al., 2022). It is represented as a combination of
the partial derivatives of these variables and their corresponding individual error terms (Ricker et al., 2014):

$$\sigma_{\rho_i} = \sqrt{(1 - \frac{h_f}{h_i} + \frac{h_s}{h_i})^2 \times \sigma_{\rho_w}^2 + (-\frac{h_s}{h_i})^2 \times \sigma_{\rho_s}^2 + (\frac{\rho_w h_f - \rho_w h_s + \rho_s h_s}{h_i^2})^2 \times \sigma_{h_i}^2 + (\frac{\rho_w - \rho_s}{h_i})^2 \times \sigma_{h_s}^2 + (-\frac{\rho_w}{h_i})^2 \times \sigma_{h_f}^2}. \quad (6)$$

where $\sigma_{\rho_w}$ is the seawater density uncertainty, $\sigma_{\rho_s}$ is the snow bulk density uncertainty, $\sigma_{h_i}$ is the sea ice thickness uncertainty, $\sigma_{h_s}$ is the snow depth uncertainty, and $\sigma_{h_f}$ is the total freeboard uncertainty.

In this study, $\sigma_{\rho_w}$ was set to 0.5 kg m−3 according to Wadhams et al. (1992), and $\sigma_{\rho_s}$ was set to half the 95% confidence
interval of the fitted function for snow bulk density (see blue shaded area in Fig. 4c). $\sigma_{h_i}$ and $\sigma_{h_s}$ were both set to 0.02 m following Lei et al. (2022). $\sigma_{h_f}$ was regarded as the IS2 modal freeboard uncertainty ($\sigma_{mode}$) (Eq. 3). It should be noted that the IBD uncertainty ($\sigma_{\rho_i}$) calculated here represents the total random uncertainty. The IBD estimates with uncertainties greater than 40 kg m$^{-3}$ (~4.5%) were excluded from subsequent analyses to ensure data quality.

### 2.2.4 Parameterization principles for IBD

Unlike previous studies that have focused solely on the potential relationships between IBD and univariate sea ice parameters, such as sea ice thickness and freeboard (Kovacs, 1997; Jutila et al., 2022), this study additionally examined two bivariate parameters: ice draft-to-thickness ratio and ice freeboard-to-total freeboard ratio, as they may indirectly indicate sea ice stratification. In addition, the univariate sea ice parameters selected in this study included sea ice thickness, total freeboard, sea ice draft, and sea ice freeboard. The potential relationships between IBD (i.e., SI) and each MOSAiC ice
parameter were explored using linear and non-linear regression analyses. The training dataset for the regression model spanned the ice growing season (October to April) and incorporated measurements from all 15 deployed buoys within a 50 km radius, ensuring good spatial and temporal representativeness.

## 3 Results

### 3.1 Integrated observations of sea ice and snow during MOSAiC

Figure 4 illustrates the seasonal fluctuations in sea ice thickness, snow depth, snow bulk density, and IS2 modal freeboard throughout the freezing season. All buoy-measured sea ice thicknesses and IS2 modal freeboards demonstrated a distinct seasonal increase, whereas snow depths at the buoy sites exhibited significant spatial heterogeneity and considerable temporal variability. Note that the buoy deployment sites included both SYI and FYI (Lei et al., 2022; Perovich et al., 2023). All selected buoys were strategically deployed on the level ice, effectively capturing thermodynamically driven ice growth



(Koo et al., 2021). In this study, we ensured at least 10 available buoys per day and used their mean sea ice thickness and snow depth as inputs to the IBD retrieval (Fig. 2).

In mid-October 2019, the initial thickness of the deployed sites varied from approximately 0.30 to 1.80 m (Fig. 4a). From mid-October to mid-April, the growth rates of sea ice thickness ranged from 0.08 to 0.31 m per month (see Table S2 for details). Additionally, exceptionally large ice growth was observed at some sites due to the formation of platelet ice or snow-

ice. Sea ice growth commenced relatively early at the I3 buoy on 10 October 2019, driven by the thinnest initial ice thickness of approximately 0.30 m. From mid-October to mid-November, remarkable growth of approximately 0.5 m was observed at this site, significantly higher than at other buoys. Between 14 and 15 January 2020, the I2 buoy recorded a notable increase in ice thickness of about 0.10 m, which was attributed to the accumulation of platelet ice at the bottom due to super-cooled water (Perovich et al., 2023). The T72 buoy recorded snow-ice formation in mid-April associated with the flooding process

and local depression caused by a storm event (Lei et al., 2022). Apart from these events, the seasonal change in ice thickness shows a relatively smooth pattern. However, we need to emphasize that these anomalous synoptic-scale processes may lead to notable uncertainties in the IBD retrieval.

Seasonal variations in snow depth derived from both buoys and transects of level ice segments demonstrated a uniform pattern during the freezing season (Fig. 4b). Throughout the observation period, buoy-derived snow depths ranged from

approximately 0.05 to 0.30 m, with an increasing trend of 0.010 m per month, which was slightly lower than the trend of 0.017 m per month for the snow transects because the level ice segments close to the deformed ice favored snow accumulation (Itkin et al., 2023). Nevertheless, the average discrepancy in snow depth between buoys and transects was slight (~0.045 m), indicating good agreement. After the strong snow drifting period (yellow arrow in Fig. 4b), the variance in snow depth between buoy sites increased significantly. Despite the relatively high spatial variability in snow depth,

comparisons with transect data, which have a relatively more comprehensive measurement coverage (Itkin et al., 2023), show that the buoy array measurements are sufficiently representative of snow depth on level ice at the MOSAiC local-scale. Therefore, we have sufficient confidence to use the buoy-derived snow depth to retrieve the IBD (Fig. 2).

Snow bulk density exhibited significant temporal variation due to fresh snowfall or snow metamorphism (Macfarlane et al., 2023) (Fig. 4c). A considerable range of snow bulk density, mainly between 200 and 350 kg m$^{-3}$, was observed during

the MOSAiC freezing season, with apparent synoptic-scale variability that may be linked to snowfall events. Overall, a statistically significant increasing trend ($P < 0.001$) of 11 kg m$^{-3}$ per month in snow bulk density was observed, attributed to increasing grain size and snow stratification (Wagner et al., 2022). To extend the temporal availability of snow bulk density data, we used the fitted snow bulk density estimates as input to the IBD retrieval (Fig. 2).

The seasonal variation of the IS2 modal freeboard is shown in Fig. 4d. A short data gap occurred from February to early

March as the MOSAiC ice floes drifted north of 88.5° N, beyond the spatial coverage of the IS2 measurements. During the initial phase of ice growth, the IS2 modal freeboard increased from 0.16 m in mid-October 2019 to 0.42 m in mid-April 2020, with a growth rate of 0.037 m per month ($P < 0.001$). Episodic storm events, often associated with ice field deformation, have no discernible effect on the change in IS2 modal freeboard. The observed relatively smooth and significant linear trend



in this parameter indicates that the primary factor influencing its change is the thermodynamic growth of sea ice. It has also been reported that the IS2 modal freeboard closely aligns with the modal freeboard obtained from airborne laser scanning within 100 km of the MOSAiC CO (Hutter et al., 2023), further enhancing our confidence in using the IS2 modal freeboard to retrieve the IBD for the level ice at the MOSAiC local-scale (Fig. 2).

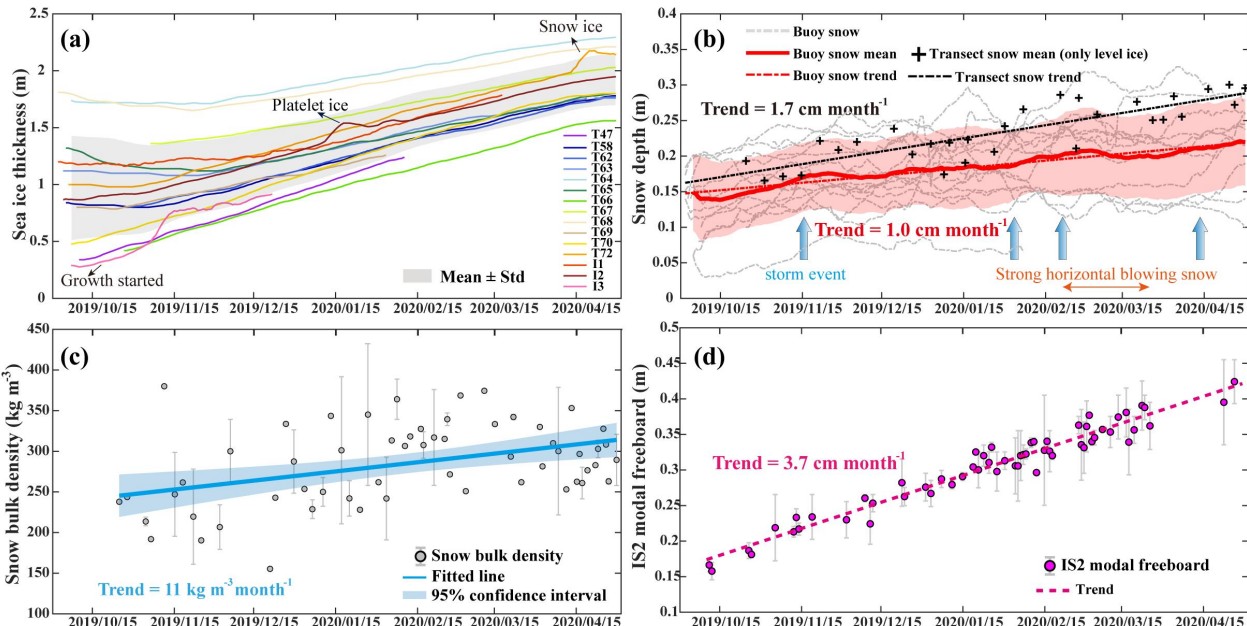

**Figure 4.** Variations of sea ice and snow during the MOSAiC freezing season. **(a)** Sea ice thickness measurements from the IMBs. The peculiarities of T72, I2, and I3 sites (as described in the text) are indicated by black arrows. The gray shaded band indicates the mean ± standard deviation (at least 10 buoys). **(b)** Snow depth measurements obtained from buoys and transects. The blue arrows mark four storm events, and the yellow arrow indicates a period characterized by a strong snow drifting event, according to Wagner et al. (2022). **(c)** Snow bulk density derived from snow pit measurements. **(d)** Seasonal evolution of the IS2 modal freeboard, with uncertainties indicated by gray error bands.

### 3.2 Seasonal evolution of IBD during the freezing season

We found that the SI underwent a statistically significant decreasing period (~16 kg m$^{-3}$ per month) from mid-October to mid-January and then transitioned to a relatively stable phase (~897 ± 11 kg m$^{-3}$) that lasted until mid-April (Fig. 5). The notable decreasing trend in the early ice growth season can be attributed to desalination processes (Timco and Frederking, 1996; Hutchings et al., 2015; Pustogvar and Kulyakhtin, 2016; Petrich and Eicken, 2017) (see Section 4.2 for details). Furthermore, there is evidence that a warm air intrusion event occurred during the MOSAiC expedition in mid-November 2019, which significantly increased sea ice temperature and permeability and facilitated sustained brine discharge (Angelopoulos et al., 2022).





By comparing the SI (mixed FYI and SYI) with ice densities derived from the Arctic Sever expedition (Sever, 1980–1989), in situ measurements (average of FYI and MYI, 2000–2015), and IceBird multi-sensor measurements (April 2017

and 2019), we observed generally good agreement from January to April within the uncertainty range of the SI (Fig. 5). In the autumn, the upper limit of the A10 FYI density is in agreement with the SI, while in the spring a better agreement with the lower limit of the A10 is observed. The seasonal difference highlights the high uncertainty of using the climatological IBD (e.g., A10) to retrieve ice thickness from space, as they fail to account for a significant seasonal evolution of IBD.

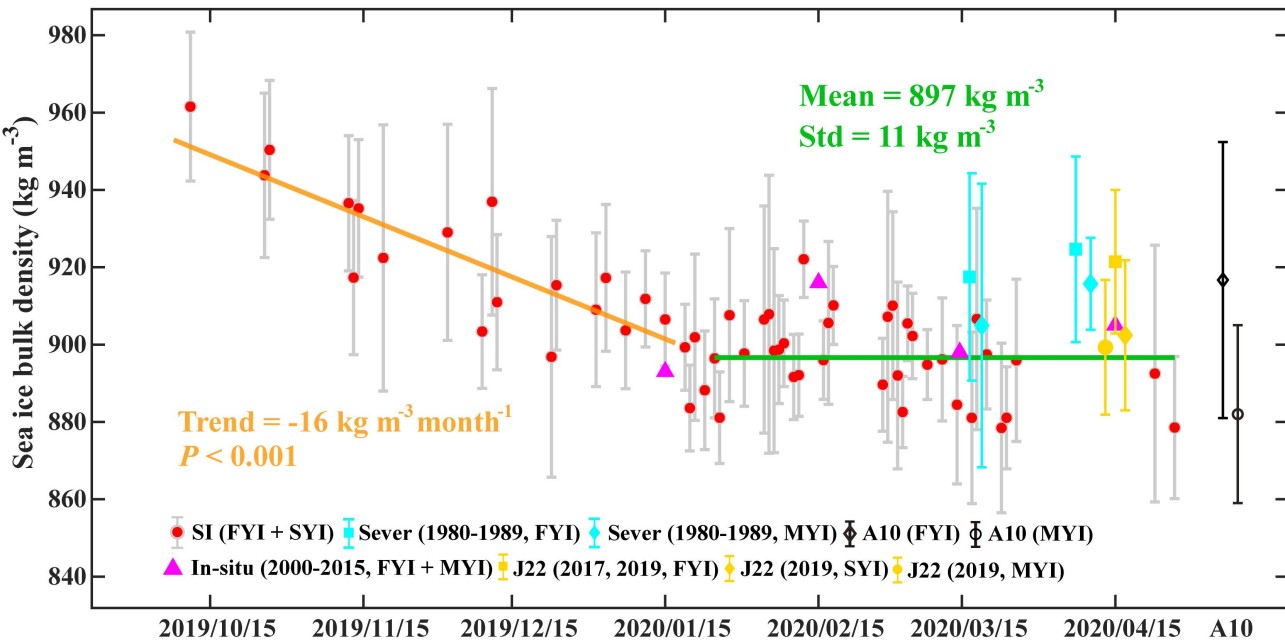

**Figure 5.** Seasonal evolution of SI during the MOSAiC freezing season. The orange line indicates a significant decreasing trend in SI, while the green line indicates the mean value of SI during a relatively stable phase. Also shown are the mean IBDs of FYI (black diamond) and MYI (black circle) from the A10 climatology (Alexandrov et al., 2010), the mean IBDs of FYI (cyan square) and MYI (diamond) estimated during the Arctic Sever expedition from 1980 to 1989 (Shi et al., 2023), the mean ice densities (FYI and MYI, purple triangle) from 2000 to 2015 based on in situ observations (Ji et al., 2021), and the mean IBDs of FYI (yellow square), SYI (diamond), and MYI

(circle) based on AWI Icebird multi-sensor measurements in April 2017 and 2019 (Jutila et al., 2022). The gray error bars indicate the uncertainty of the SI, and the other error bars indicate one standard deviation. Note that the density data from the historical measurements correspond to the month, regardless of the year.

The SI exhibited greater variability than the core-based IBDs (Fig. 6), the latter representing daily averages of all samples

from the MOSAiC CO floe and 3 DN sites (L1, L2, and L3), underscoring an inherent difference between the local-scale IBDs and those obtained from sporadic sampling sites. Interestingly, the core-based IBDs for FYI and SYI displayed a seasonal variation pattern similar to the SI, characterized by a significant decrease followed by stabilization. However, in the core-based measurements, the magnitude of the reduction in the early period was less pronounced, and the transition to the





relatively stable phase occurred earlier than in the SI. Specifically, the IBDs of the SYI cores remained consistently lower

than those of the FYI cores, with a statistically significant decreasing trend from early October to mid-November (~5.7 kg m$^{-3}$ per month, $P < 0.05$). From mid-November to mid-April, the mean IBD (± standard deviation) for the SYI was ~922.2 ± 1.1 kg m$^{-3}$. For the FYI cores, a decreasing trend of ~2.8 kg m$^{-3}$ ($P < 0.05$) in IBD was observed from early November to early December, with a subsequent mean of ~924.7 ± 0.7 kg m$^{-3}$ from early December to mid-April.

As previously highlighted by Jutila et al. (2022), IBD can exhibit significant spatial heterogeneity, with values ranging

from 800 to 1000 kg m$^{-3}$ across the airborne survey track (~40 m spatial resolution). Similarly, the IBDs determined in this study also exhibited comparable heterogeneity (detailed in Section 4.3). We also acknowledge that certain synoptic events may have influenced the parameters used to estimate the IBD, potentially affecting the results at specific sites. Nevertheless, our analysis suggests that the seasonal evolution of IBD is evident for both the local mean and sporadic sampling site values.

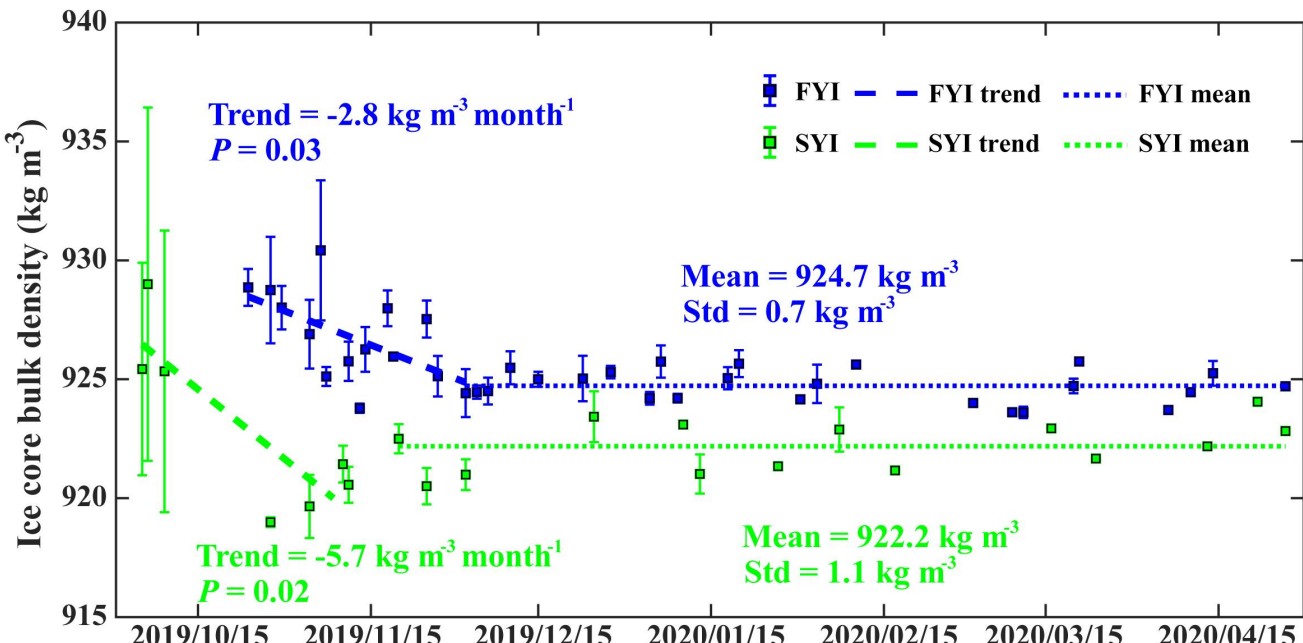

**Figure 6.** Seasonal evolution of core-based IBDs for FYI and SYI, respectively, during the MOSAiC freezing season. The blue or green dashed line indicates a significant decreasing trend, and the dotted line indicates the mean IBD during the relatively stable phase.

### 3.3 Parameterization of IBD

This section explores the potential relationships between local-scale IBD (i.e., SI) and other MOSAiC sea ice parameters to develop updated parametrizations for IBD (Fig. 7). Overall, the results revealed that all selected univariate parameters

displayed a significant quadratic non-linear correlation with IBD. In contrast, the multivariate parameters exhibited a robust linear relationship. Notably, the ice draft-to-thickness ratio demonstrated the optimal fitting performance, significantly surpassing the individual ice parameters and the ice freeboard-to-total freeboard ratio. The IBD increases with increasing ice



draft-thickness ratio, with $R^2$ and $RMSE$ values of 0.99 ($P < 0.001$) and 1.62 kg m$^{-3}$, respectively. For the other bivariate parameter, the ice freeboard-to-total freeboard ratio also performed well, exceeding all univariate parameters, with $R^2$ and

$RMSE$ values of 0.93 ($P < 0.001$) and 4.76 kg m$^{-3}$, respectively. The strong linear relationship between the two bivariate parameters and IBD further indicates that the relative proportions of the above- and below-water components of sea ice, as well as the relative proportions of snow cover and ice freeboard, are potentially significant indicators of IBD, supporting previous suggestions (Alexandrov et al., 2010; Shi et al., 2023).

Among the univariate parameters (Fig. 7a-d), sea ice freeboard showed the optimal fitting performance ($R^2 = 0.89$ and

$RMSE = 6.14$ kg m$^{-3}$), and its fitted line indicated that IBD decreases with increasing sea ice freeboard, in close agreement with Jutila et al. (2022) (hereafter J22). Here, we note that sea ice freeboard can be considered as a pivotal and robust parameter in determining IBD, both in late spring (J22) and throughout the freezing season (this study). The fitting performance of total freeboard against IBD ($R^2 = 0.85$ and $RMSE = 7.30$ kg m$^{-3}$) was relatively less effective than that of sea ice freeboard. Moreover, the ice thickness and draft both exhibited relatively poor fitting results, with $R^2$ values of 0.65 and

0.61, respectively. Nevertheless, these parameters retain potential as viable options for IBD parameterizations, particularly when other ice parameters are not readily available.

Following the regression analyses, updated IBD parameterizations were developed as shown in Eqs. (7) and (8). These parameterizations are anticipated to be applicable throughout the freezing season in regions encompassing both the FYI and SYI, spanning a spatial extent of tens of kilometers, which aligns with the grid scales employed by most satellite products

and numerical models. We emphasize that the range for each parameter input induced by the constrained training dataset was included (Table 1). However, we also argue that the range of ice thickness (0.9−1.9 m) is particularly representative of first- or second-year level ice. Therefore, the parameterization scheme provided in this study is applicable to these two sea ice types, which are also the dominant ice types for the current Arctic Ocean. For specific applications, it is recommended to select the appropriate parameterized equation or combination based on the available sea ice parameters. In addition, these

parameterized equations exclude time-adjusted regression coefficients due to the lack of sufficient sampling at the seasonal or sub-seasonal timescale.




**Figure 7.** Parameterization of the IBD, including regression models using **(a)** sea ice thickness, **(b)** total freeboard, **(c)** sea ice draft, **(d)** sea ice freeboard, **(e)** ice draft-to-thickness ratio, and **(f)** ice freeboard-to-total freeboard ratio. Each panel shows model fit metrics, including the coefficient of determination ($R^2$) and the root mean square error ($RMSE$). The red shaded bands indicate the 95% confidence intervals of the fitted curves. The number of samples used to train the model is 55. Note that the statistical $P$-value for all results is less than 0.001 and the unit of $RMSE$ is in kg m$^{-3}$.



The updated parameterized equations are as follows:

- Univariate parameterized equations (kg m$^{-3}$):

$$\overline{\rho_i(x,y,t)} = a1 \times \overline{X(x,y,t)}^2 + a2 \times \overline{X(x,y,t)} + a3, \tag{7}$$

• Bivariate parameterized equations (kg m$^{-3}$):

$$\overline{\rho_i(x,y,t)} = a1 \times \overline{X(x,y,t)} + a2. \tag{8}$$

where the $\overline{\rho_i(x,y,t)}$ and $\overline{X(x,y,t)}$ represent the local-scale IBD and other ice parameters at any given location $(x,y)$ and time $(t)$, respectively. Table 1 lists the regression coefficients for Eqs. (7) and (8). It should be noted that these parameterized equations are not independent.


**Table 1.** Regression coefficients of the parameterized equations (kg m$^{-3}$). The uncertainties of the regression coefficients are given in brackets.

| $\overline{X(x,y,t)}$ | Variable type | a1 | a2 | a3 | Input range in this study |
|---|---|---|---|---|---|
| Sea ice thickness (m) | Univariate | 69 (43) | −246 (121) | 1112 (83) | 0.9−1.9 m |
| Total freeboard (m) | Univariate | 794 (485) | −794 (287) | 1058 (42) | 0.16−0.42 m |
| Sea ice draft (m) | Univariate | 108 (66) | −338 (170) | 1158 (108) | 0.9−1.7 m |
| Sea ice freeboard (m) | Univariate | 1529 (769) | −737 (176) | 968 (10) | 0.03−0.20 m |
| Ice draft-to-thickness ratio | Bivariate | 954 (23) | 27 (21) | — | 0.89−0.97 |
| Ice freeboard-to-total freeboard ratio | Bivariate | −217 (16) | 986 (6) | — | 0.15−0.50 |

## 4 Discussion

### 4.1 Uncertainties and limitations of IBD retrieval

Recognizing the complexities of matching the IS2 along-track freeboards with the MOSAiC DN measurements is imperative. A primary challenge is the temporal discrepancies between the satellite and buoy measurements, typically ranging from several hours to half a day. Another challenge concerns the limited IS2 track coverage around the MOSAiC CO (Fig. 3). In addition, the retrieval of IBD relied on snow pit measurements, subject to spatial and temporal sampling limitations. The buoy-derived snow depth or ice thickness has an uncertainty of ~2 cm, which also contributes to the uncertainty in the IBD

retrieval. On this basis, we further quantified the relative contribution (RC) of the different input parameters used in Eq. (5) to the total uncertainty in the IBD (see details in Text S1). Throughout the study period, snow depth and total freeboard (i.e., IS2 modal freeboard) contributed the most to the total uncertainty in the IBD, with mean RCs of 47% and 50%, respectively.





In contrast, the mean RCs of seawater density (0.1%), snow bulk density (1.7%), and sea ice thickness (1.2%) were considered negligible. This suggests that snow depth and total freeboard are the most sensitive factors in estimating IBD,
which require more representative and accurate measurements.

In terms of interpretation, the local IBD results (SI) mainly reflect level ice mixed with FYI and SYI, which lack information on different ice ages and types (especially deformed ice) and additionally contribute to the potential limitations and uncertainties of the parameterization scheme. Despite these limitations, we argue that the optimized parameterization scheme derived from the MOSAiC observational data demonstrates broad applicability. The MOSAiC DN region is
primarily characterized by SYI, with only a tiny portion comprising FYI (Krumpen et al., 2020). This situation represents the dominant pattern of the marginal ice field during the early ice-growing season, constituting approximately 30−40% of the total Arctic ice area. Furthermore, level ice is consistently regarded as the primary form of sea ice in the hydrostatic equilibrium calculation for the ice field (Sumata et al., 2023). In order to further develop parameterization schemes for the pan-Arctic Ocean, we also recommend expanding the observational data from the MYI regions.

**4.2 Understanding the decreasing trend in IBD**

During the freezing season, the pore spaces of sea ice initially saturated with saline gradually become filled with air content, thus reducing IBD (Petrich and Eicken, 2017). Additionally, air temperature is recognized as a potential factor that influences IBD by increasing ice temperature. In particular, warm air intrusion events can significantly enhance the efficiency of sea ice brine drainage, provided these events persist for an adequate duration (Angelopoulos et al., 2022).

Timco and Frederking (1996) found significant differences in IBD between the sections above and below the waterline. Specifically, IBDs above the waterline varied considerably, ranging from 840 to 910 kg m$^{-3}$ for FYI and 720 to 910 kg m$^{-3}$ for MYI. Below the waterline, however, IBDs are much more uniform, ranging from 900 to 940 kg m$^{-3}$ for both FYI and MYI. Furthermore, Pustogvar and Kulyakhtin (2016) demonstrated that air porosity significantly influences IBDs, while the effects of salinity and temperature are minor, which explains the considerable variation in reported IBDs despite low
estimates of brine discharge (Timco and Frederking, 1996).

Considering these factors, the seasonal evolution of IBD during the freezing season is anticipated to be highly complex, resulting from a combination of internal properties and external factors. However, its significant decreasing trend from autumn to early spring appears to be closely related to brine drainage and increasing air content as sea ice ages. Our improved understanding of the declining trend in IBD during the early freezing season and its implications for seasonal
variations in remotely sensed ice thickness retrievals provides a critical perspective on the spatio-temporal adaptability of IBD parameterization.

**4.3 Spatial heterogeneity of IBD across the MOSAiC ice floes**

The spatial heterogeneity of IBD across the MOSAiC ice floes was also investigated. The IBD retrieval procedure described in Fig. 2 was partially modified to estimate IBD for each buoy site. These modifications included (1) employing ice



thickness and snow depth measurements obtained at each buoy site and (2) specifying a daily time window and a 20-km search radius from each buoy site to estimate the IS2 modal freeboard, as in Koo et al. (2021). However, it is crucial to acknowledge the significant spatial scale differences that occur when matching individual buoys to satellite along-track measurements. Therefore, our goal was not to target the relative accuracy of IBDs derived from a single buoy site. Instead, we aimed to elucidate the mean variance in IBD across buoy sites throughout the freezing season, thereby highlighting

broader spatial variations.

Figure 8 shows the derived IS2 modal freeboard at each buoy site. To ensure sufficient IS2 samples for a log-normal fit, the IS2 modal freeboard was only calculated when the IS2 along-track freeboard segments were greater than 4500 within the 20 km search radius. Overall, the IS2 modal freeboards for all buoy sites showed a uniform increasing pattern but with some differences in their trends and means. The trends in IS2 modal freeboard ranged from ~3.18 to 4.14 cm per month, with the

lowest recorded at the T47 buoy and the highest at the T69 buoy. The mean values of IS2 modal freeboard were relatively comparable for the different buoy sites throughout the growth season, ranging from ~0.27 to 0.32 m. These findings indicate that the magnitude of thermodynamically driven sea ice growth exhibits significant spatial variability, possibly related to initial ice thickness, sea ice types, and snow accumulation.

Based on the IS2 modal freeboard, snow depth, sea ice thickness obtained at each buoy site, and the fitted snow bulk

density, we calculated the IBDs for the different buoy sites, as depicted in Fig. 9. It should be emphasized that the retrieved IBDs represent the entire level ice layer surrounding each buoy site, possibly mixing FYI and SYI. To ensure the reliability and representativeness of the results, we retained the buoy data that contained at least 20 valid density records throughout the freezing season. Moreover, the IBD uncertainty for any given day was maintained below 40 kg m$^{-3}$. We observed significant differences in IBD across the buoy sites (Fig. 9), which ranged from approximately 800 to 1000 kg m$^{-3}$ and align well with

the J22 results. The T64 buoy, with an initial ice thickness of 1.74 m, exhibited the highest density (939 kg m$^{-3}$). In contrast, the T70 buoy, with an initial ice thickness of 0.48 m, had the lowest (861 kg m$^{-3}$), with a significant difference in mean IBD of up to 78 kg m$^{-3}$. Thus, this disparity could be related to initial ice thickness. These findings underscore the significant spatial heterogeneity in IBD, highlighting further challenges for relatively small-scale, satellite track-based sea ice thickness retrievals.






**Figure 8.** IS2 modal freeboards for different buoy sites. Each panel shows the mean, standard deviation, and trend ($P < 0.001$). The error bars represent the uncertainties in the modal freeboards, and the red dotted lines indicate the linear trends.





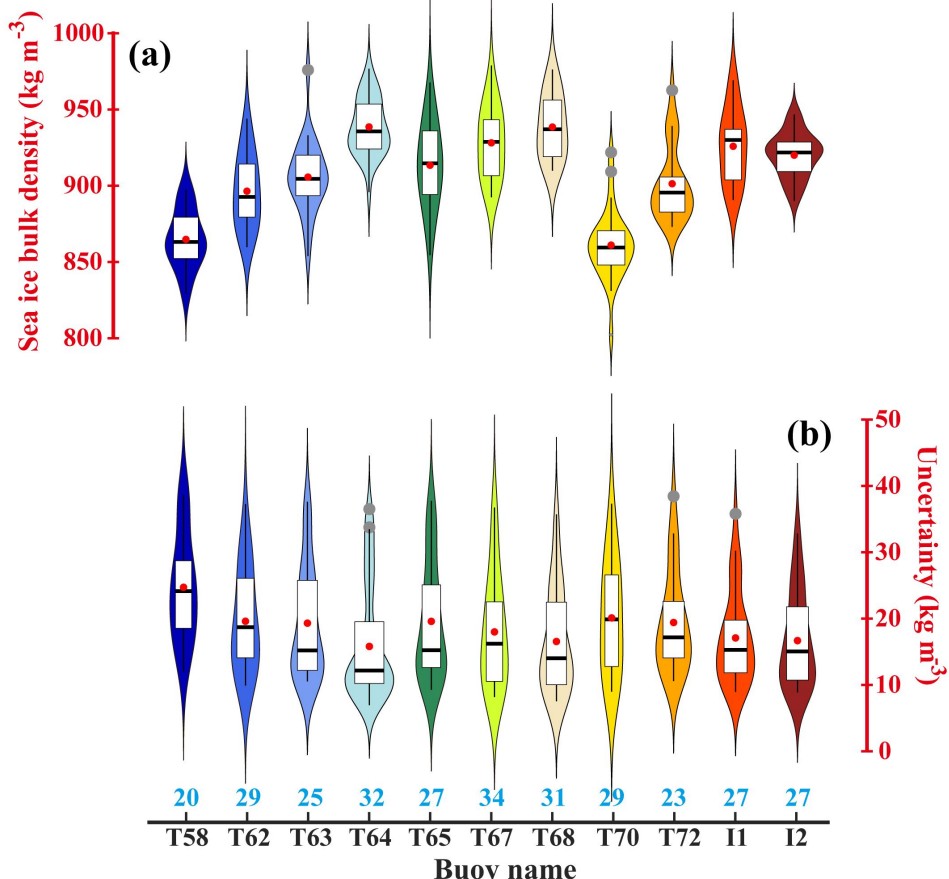

**Figure 9. (a)** IBD and its **(b)** uncertainty for different buoy sites. The box plots illustrate the interquartile range, with the median marked by a black line, the mean by a red dot, and anomalies by gray dots. The colored bands around the boxes indicate the probability density distribution of IBD. The blue numbers denote the effective IBD records for each buoy site during the freezing season.

### 4.4 Applicability of the parameterization based on ice draft-to-thickness ratio to deformed, rough, and older ice

The ice draft-to-thickness ratio exhibited the optimal fit performance for characterizing IBD (Fig. 7e). However, as mentioned above, the applicability of this parameterization is expected to be primarily limited to the level ice due to inherent limitations in the training data. This section examines whether the ice draft-to-thickness ratio is still strongly correlated with IBD for deformed, rough, and older sea ice to explore its broader applicability.

Here, the AWI IceBird multi-sensor sea ice data provided by Jutila et al. (2022) were used for further analysis. This dataset includes total thickness, sea ice thickness, snow depth, total freeboard, ice type, IBD, and their associated uncertainties. The data were collected in April 2017 and 2019 along a 3000 km profile in the western Arctic Ocean, primarily featuring MYI or SYI and including both level and deformed ice at a high spatial resolution of ~40 m. The IBD



calculations were conducted using the hydrostatic equilibrium method across various sea ice types, including FYI, SYI, and
MYI (Jutila et al., 2021a; Jutila et al., 2021b). An exponentially parameterized equation for IBD as a function of ice
freeboard (J22) was developed using this dataset. The parameterization using the ice draft-to-thickness ratio based on this
dataset was tested here to assess its applicability to more diverse ice types.

Employing the inverse distance weighting method outlined in J22, sea ice parameters were extracted at different spatial
scales (40 m, 800 m, 12.5 km, and 25 km). The results were only retained if the IBD uncertainties were less than 20 kg m$^{-3}$
to ensure data reliability. Figure 10 illustrates the fitting performance of the ice draft-to-thickness ratio against IBD at
different spatial scales. Encouragingly, a robust correlation between the ice draft-to-thickness ratio and IBD was observed at
all spatial scales, encompassing both deformed and level ice, as well as ice of varying ages (Table S3). For all ice regimes
combined, the $R^2$ of the ice draft-to-thickness ratio versus IBD exceeded 0.6 at all spatial scales, with the 40-m scale
performing best ($R^2 = 0.92$, $RMSE = 8$ kg m$^{-3}$), indicating that this parameter can be used to quantify IBD at more minor
scales. These results underscore the robustness and reliability of utilizing the ice draft-to-thickness ratio for parameterizing
IBD, further confirming the effectiveness and high confidence of our updated parameterizations.

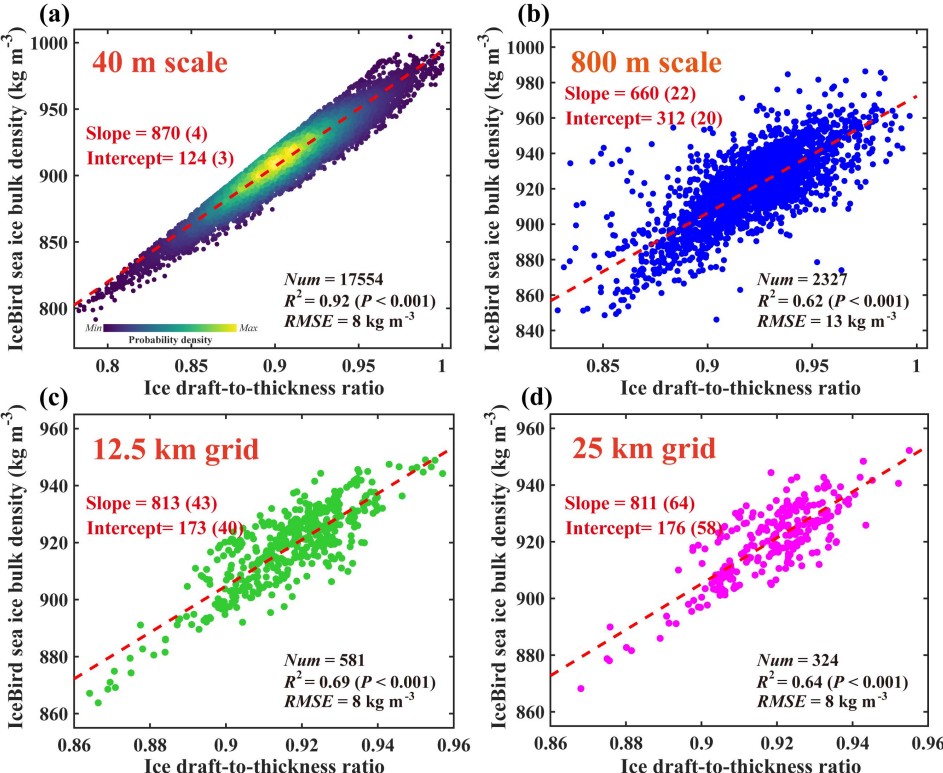

**Figure 10.** Fitting performance of the ice draft-to-thickness ratio to the IBD at different spatial scales based on AWI IceBird
multi-sensor sea ice data, including the results from **(a)** 40 m, **(b)** 800 m, **(c)** 12.5 km, and **(d)** 25 km. The red dashed line
indicates the fitted line, with the uncertainties of the slopes and intercepts in parentheses.



Furthermore, given that this dataset encompasses a range of sea ice conditions in April, we suggest that the derived regression coefficients be utilized in subsequent studies to estimate IBD for both FYI and MYI in late spring. We also recognize that the specific fitting parameters need to be adjusted according to region, sea ice conditions, and season. Therefore, extensive observational data are also needed to establish parameterization schemes with improved applicability. The spatial representativeness of buoy array measurements, which is currently limited to level ice, could be significantly improved by optimizing the deployment strategy. Specifically, by increasing buoy placements over ice ridges, refrozen melt ponds, new ice over leads, and MYI, the observations from the buoys can more accurately represent both the growth and decay phases of sea ice, as well as the accumulation and melt of snow on its surface. These additional observations could subsequently refine the parameterization of IBD.

## 4.5 Outlooks

IBD is a critical parameter that determines ice porosity (Cox and Weeks, 1983) and is the fundamental input to both thermodynamic (Ono, 1967) and mechanical (Wang et al., 2021) parameterizations of sea ice. Thus, it is of great importance for the numerical simulation of sea ice thermodynamics and dynamics, as well as for the numerical study of interactions between sea ice and ocean structures. In addition, IBD and porosity are key determinants of sea ice permeability (Golden et al., 1998), influence summer meltwater infiltration (Perovich et al., 2021), and shape the perennial greenhouse gas flux through the ice layer (Angelopoulos et al., 2022), thereby altering the stratification and biogeochemical cycling of the upper Arctic Ocean. However, the limited availability of sampling data severely hampers our understanding of IBD, resulting in a significant knowledge gap for satellite retrievals and other applications. Thus, the new approach proposed in this study to determine IBD at the basin scale using satellite altimetry data can support interdisciplinary studies.

As the most comprehensive field data on sea ice and snow are currently still from the MOSAiC observations, our research is focused on this to ensure the accuracy of the IBD results and the innovation of our methods. For alternative observational programmes, obtaining such comprehensive sea ice and snow observations at both spatial and temporal scales can be challenging. These programmes are often limited to the use of single or a few IMBs, rather than IMB arrays, which poses significant challenges in matching with satellite observations. We will continue to explore ways to use and merge broader data sources in future research to facilitate applications at the pan-Arctic scale.

## 5 Conclusion

This study provides the first estimates of IBD during the Arctic freezing season using synergistic measurements from the MOSAiC expedition and IS2/ATLAS while developing updated parameterizations for IBD. Our methodology has proven effective in determining IBD over scales of tens of kilometers, thus potentially improving the retrieval of ice thickness. This work will significantly contribute to the coordination of multiple altimetry missions that measure ice thickness, including CryoSat-2, ICESat-2, and the upcoming CRISTAL mission (Kern et al., 2020). Given the rapid change in the composition of



Arctic sea ice from thicker, older MYI to thinner, younger seasonal ice components (Sumata et al., 2023), our updated parameterizations, explicitly designed for Arctic FYI and SYI, could serve as a valuable reference for future satellite retrieval efforts.

The following are the main findings of this study. At the MOSAiC local-scale (~50 km), the IBD demonstrated a statistically significant decreasing trend from mid-October to mid-January (~16 kg m$^{-3}$ per month), followed by a relatively stable phase lasting until mid-April (~897 ± 11 kg m$^{-3}$). The observed decreasing trend is probably related to the desalination process and increased internal porosity as the sea ice aged. The core-based IBDs displayed a comparable seasonal evolution. However, unlike the local-scale IBD estimates, the core-based IBDs varied over a smaller range and entered the relatively

stable phase earlier. This is closely related to the significant spatial heterogeneity of the IBD and the inherent deviations between the measurement and calculation methods. The seasonal variability of the IBD further underscores the high uncertainty in retrieving ice thickness using the fixed climatological IBD (A10). On this basis, we have developed updated parameterizations for the IBD via regression analyses that include univariate and bivariate ice parameters, which are anticipated to be applicable throughout the freezing season. In particular, the ice draft-to-thickness ratio was identified as the

most effective parameter for determining IBD ($R^2$ = 0.99, $RMSE$ = 1.62 kg m$^{-3}$), with the potential for application in MYI and deformed ice as well. These parameterizations hold the potential to improve the accuracy of satellite-based sea ice thickness retrievals.

*Data availability.* Snow depth and sea ice thickness data derived from SIMBA buoy measurements are available from

PANGAEA: https://doi.org/10.1594/PANGAEA.938244. Snow depth and sea ice thickness data derived from SIMB buoy measurements are available from the Arctic Data Center: https://doi.org/10.18739/A20Z70Z01. *Magnaprobe*-based snow depth data from snow transects are available from PANGAEA: https://doi.org/10.1594/PANGAEA.937781. Snow pit data collected during the MOSAiC expedition are available from PANGAEA. Core-based ice density data from eight MOSAiC sites are available from PANGAEA: https://doi.org/10.1594/PANGAEA.943811. ICESat-2 ATL10 total freeboard data

(version 5) are available from NSIDC: https://nsidc.org/data/atl10/versions/5. AWI Icebird multi-sensor sea ice data collected in April 2017 and 2019 are available from PANGAEA: https://doi.org/10.1594/PANGAEA.933883 (April 2017) and https://doi.org/10.1594/PANGAEA.933912 (April 2019). The bathymetry data used in Fig. 1 are available from the National Oceanic and Atmospheric Administration (NOAA): https://www.ngdc.noaa.gov/mgg/global/relief/ETOPO1. The local-scale sea ice bulk density and IS2 modal freeboard data retrieved in this study are available from ZENODO:

https://zenodo.org/doi/10.5281/zenodo.11055727.

*Code availability.* The procedures for IS2 modal freeboard extraction and sea ice bulk density retrieval are available from ZENODO: https://zenodo.org/doi/10.5281/zenodo.11055727.



*Author contributions.* YZ (Yi Zhou), XW and RL designed the study, developed the methodology, and performed the primary analysis. RL and DP developed the snow depth and ice thickness data for SIMBA or SIMB. LVA, YZ (Yu Zhang) and CH contributed to the interpretation of the results. YZ (Yi Zhou), XW, and RL prepared the manuscript. All authors contributed to revising the manuscript.

*Competing interests.* At least one of the (co-)authors is a member of the editorial board of *The Cryosphere*.

*Acknowledgements.* We extend our profound appreciation to our team members, collaborators, and advisors whose invaluable contributions have been pivotal to this research. The datasets utilized in this study were mainly collected from the MOSAiC expedition, an international effort. We are grateful to everyone involved in the MOSAiC 2019/2020 expedition
aboard the research vessel *Polarstern*. Special thanks are also given to NASA for providing the ICESat-2 along-track sea ice freeboard data. These data are critical for revealing the seasonal evolution of sea ice bulk density.

*Financial support.* YZ (Yi Zhou) and XW were supported by the National Natural Science Foundation of China (Grant No. 42276237) and the Oceanic Interdisciplinary Program of Shanghai Jiao Tong University (No. SL2022PT205). RL was
supported by the National Natural Science Foundation of China (Grant No. 42325604). DP was supported by the National Science Foundation (NSF-OPP-2034919 and NSF-OPP-2138785). YZ (Yu Zhang) was supported by the National Natural Science Foundation of China (Grant No. 42376231 and Grant No. 42130402).

*Disclaimer.* Publisher's note: Copernicus Publications remains neutral with regard to jurisdictional claims made in the text,
published maps, institutional affiliations, or any other geographical representation in this paper. While Copernicus Publications makes every effort to include appropriate place names, the final responsibility lies with the authors.





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
