# Peer review of "Seasonal evolution and parameterization of Arctic sea ice bulk density: results from the MOSAiC expedition and ICESat-2/ATLAS"

_EGUsphere, 2024_

## Author Comment (AC1)

**Responses to Community Comment (Dr. Arttu Jutila)**
* * *
Dear Dr. Arttu Jutila,

Thank you for your constructive comments. We will address your insights with comprehensive clarifications and revisions throughout our manuscript. We have outlined the original comments in black with our planned responses highlighted in blue. Kindly refer to the attached document.

Best regards,
Yi Zhou and other co-authors.
* * *
Dear Zhou et al.,
Dear handling editor,

The competing interests policy of The Cryosphere prohibits me from acting as an official referee for this manuscript due to recent collaborations with some of the coauthors. Therefore, I am posting this comment as a member of the scientific community to discuss some matters related to it.

In the manuscript by Zhou et al., now under peer-review process and public discussion, sea-ice bulk density is derived using the hydrostatic equilibrium equation and values of modal total freeboard from the satellite laser altimeter ICESat-2, mean snow depth and sea-ice thickness from 15 autonomous ice mass-balance buoys (IMB) deployed within the MOSAiC Distributed Network (DN), and mean snow density from snow pit measurements conducted in the MOSAiC Central Observatory (CO) from October 2019 to April 2020.

In 2022, I have authored a paper in The Cryosphere on the same general topic, sea-ice bulk density, using a similar approach but simultaneous airborne multi-sensor measurements from the AWI IceBird program:

Jutila, A., Hendricks, S., Ricker, R., von Albedyll, L., Krumpen, T., and Haas, C.: Retrieval and parameterisation of sea-ice bulk density from airborne multi-sensor measurements, The Cryosphere, 16, 259–275, https://doi.org/10.5194/tc-16-259-2022, 2022.

This work is referenced many times and data originating from this study are used in the manuscript by Zhou et al.

First, I want to inform that there is a recently published new version of the AWI IceBird airborne sea-ice parameter dataset. In the new version, the quality flag identifying level and deformed ice has been rectified.

Jutila, A., Hendricks, S., Ricker, R., von Albedyll, L., and Haas, C.: Airborne sea ice parameters during the IceBird Winter 2019 campaign in the Arctic Ocean, Version 2, https://doi.org/10.1594/PANGAEA.966057, 2024.

Jutila, A., Hendricks, S., Ricker, R., von Albedyll, L., and Haas, C.: Airborne sea ice parameters during the PAMARCMIP2017 campaign in the Arctic Ocean, Version 2, https://doi.org/10.1594/PANGAEA.966009, 2024.

**Authors' response:** We appreciate your clarification. We will incorporate the latest version of the AWI IceBird dataset and describe it in Section 2.1.

Regarding the study of Zhou et al., I would like to raise general concerns and perhaps some misunderstandings of my paper. The general points are the following:

**Spatial scales.** While the presented study broadens the knowledge with the aspect of seasonal evolution of remotely sensed sea-ice bulk density, I am concerned about the different magnitudes of spatial scales utilized in the derivation. More specifically, you use total freeboard from the ICESat-2 satellite laser altimeter orbits extracted within a circle around the CO that has a diameter of 100 km; snow depth and sea-ice thickness derived from 15 autonomous IMBs in the DN within circle around the CO that has a diameter of 70 km (in the beginning of the drift, but how about later after being affected by sea-ice dynamics for months?) while the data are inherently point measurements; and snow density derived from snow pit measurements within the CO that extends over an area with a diameter of only few hundred meters while the data are inherently point measurements. None of these data sources have real spatial overlap with each other. This effectively diminishes the study to use ice-type-averaged values (not far from Alexandrov et al.'s (2010) multi-year ice density derivation with climatological values from literature) as the measurements are not from the same piece of ice - not even remotely.

**Authors' response:** Thank you for your insightful comments on spatial scales. We will add the discussion on the uncertainties involved in matching datasets across different spatial scales in the revised manuscript. *However, there are some misconceptions about spatial scales that we need to clarify.*

**Buoy deployment sites (relatively stable)**
To begin with, we would like to clarify that the radius used for extracting **modal freeboard** data from ICESat-2 (IS2) orbits around the Central Observatory (CO) was **50 km**, **not 100 km**. Furthermore, the autonomous Ice Mass Balance Buoys (**IMBs**) in the Distributed Network (DN) provided measurements within a radius of **30-40 km** from the CO, **rather than 70 km**, and none of them are in the wider coverage of *Extended Network* (Krumpen et al., 2020; Lei et al., 2022; Rabe et al., 2024). It is also important to note that our main study period covers the freezing season from October to April. During this period, the buoy deployment areas are less influenced by sea ice dynamics, which significantly intensify from early summer onwards (Krumpen et al., 2021; von Albedyll et al., 2022). All selected buoys were located within a 50 km radius of the CO throughout the study period, as shown in Fig. A1.

[Figure]

**Fig. A1.** Distance of all selected buoy sites from the CO during the MOSAiC freezing season.

**Spatial representation, not spatial overlap**

We acknowledge that the measurements from the IMB array, the IS2 tracks, and the snow bulk density data do not spatially overlap. Instead, our study emphasizes spatial representation over direct overlap for the study region of kilometers to tens of kilometers. Given the challenges of acquiring multi-sensor data with sufficient temporal coverage during the freezing season, our approach combines MOSAiC and IS2 data to track the seasonal evolution of sea ice bulk density. This method aims to reasonably represent the average conditions of the MOSAiC ice floes at the 50 km scale (or DN scale), focusing on the representativeness of data, considering the spatial heterogeneity of ice thickness and snow thickness, rather than their spatial concurrence.

**Spatial compatibility of the buoy array and IS2**

Despite the IMB array consisting solely of point measurements from 15 buoys, of which at least 10 are operational continuously, we are confident in their ability to effectively represent the sea ice thickness and snow depth for level ice across the MOSAiC DN scale. Supporting evidence from numerous recent studies within the MOSAiC expedition underscores the robustness of our buoy deployment strategy. These studies showed good agreement between buoy data and more extensive airborne or transect measurements, which capture a broader range of observations (Koo et al., 2021; Krumpen et al., 2021; Rabe et al., 2024; von Albedyll et al., 2022).

We adopted a modal approach to analyze IS2 orbit data within a 50 km radius around the CO, focusing on the log-normal distribution of Arctic sea ice freeboard, well-supported by extensive research (Farrell et al., 2011; Haas, 2010; Hansen et al., 2013; Koo et al., 2021; Landy et al., 2019; Petty et al., 2016; Ricker et al., 2015; Tian et al., 2020; von Albedyll et al., 2022). The mode, representing the most common level ice freeboard and excluding deformed ice (at least statistically significant), correlates closely with thermodynamic ice growth (Koo et al., 2021; Ricker et al., 2015), providing a reliable level ice estimate within the MOSAiC DN. Despite spatial and temporal variabilities, our method uses over 15,000 daily freeboard segments, ensuring reliable and statistically robust results.

**Expected phenomena**

More importantly, we have identified a statistically significant decreasing trend in sea ice bulk density during the early freezing season (Fig. 5), thus providing some evidence for the seasonal evolution of sea ice bulk density, which is consistent with both the expected phenomenon (i.e., desalination processes) and the results reported in previous studies (Hutchings et al., 2015; Petrich and Eicken, 2017; Pustogvar and Kulyakhtin, 2016; Timco and Frederking, 1996).

**Snow bulk density from the MOSAiC CO**
All snow pit measurements in our study were conducted on level ice, excluding the period when ice ridges were sampled (Macfarlane et al., 2023). Despite spatial heterogeneity and the resulting uncertainties, our analysis shows that the data from a single floe reliably represent the snow's metamorphic processes on level ice, essential for understanding changes in sea ice bulk density during the freezing season (see Fig. 4c). We also acknowledge that observations of snow pits originating from individual ice floe (MOSAiC CO) may limit spatial representativeness. However, based on the results of on-site investigations of several ice floes at the MOSAiC DN scale, at least at the initial stage, the snow depth of MOSAiC CO, relative to other ice floes, has no special abnormal characteristics, and can be    considered representative for the snow state at the MOSAiC DN scale.

Leaving aside the issue of spatial scale, we have also quantified the uncertainties related to various input parameters in the calculation of sea ice bulk density. The relative contribution of snow bulk density to total uncertainty is around 1.7%, significantly less than that from total freeboard (50%) and snow depth (47%). Also, the estimated uncertainty in snow bulk density is about 20-30 kg m$^{-3}$, close to the representative value of 34 kg m$^{-3}$ (King et al., 2020).

In addition, using the term "local-scale" with data originating extending 100 km, when local is generally understood as < 1 km, definitely raised my eyebrows throughout the manuscript.
**Authors' response:** Defining a study area of tens of kilometers as a local scale may not be reasonable. We will revise this terminology to ensure that the expression is not misleading.

Why was ICESat-2 ATL10 rev5 used when rev6 is available? Actually, why not use the publicly available, more local MOSAiC helicopterborne laser scanner data by Hutter et al. (2023), which you also cite in the manuscript? I think that could be a feasible option to explore and it would back up better the local-scale aspect of this study. At this point, however, I must point out that I was involved in collecting and processing said data, too.
**Authors' response:** Indeed, when we finished the analysis, ATL10 V6 had not been released. Now, with some updated data processing, we have found that the updates in rev6 have minimal impact on our modal freeboard estimates, with changes of around 1-3%. This slight variation does not significantly impact the sea ice bulk density results. We commit to updating the ice bulk density to ATL10 rev6 for the final version to incorporate these improvements and ensure the most accurate data representation.
https://nsidc.org/sites/default/files/documents/user-guide/atl10-v006-userguide.pdf

As mentioned above, we focus on matching the buoy array (a sufficient number of IMBs) with all available freeboard data within the MOSAiC DN range. We considered using airborne data,

but found it insufficient—only valid observations from about ten days were available, offering limited insights into seasonal variations due to inadequate statistical sampling. Additionally, while there are more airborne records for the CO floe, the corresponding IMB data are too sparse for reliable spatial matching and confident bulk density calculations at this scale. In the revision, we will combine thses airborne data to consolidate our conclusion.

Level ice. You state that the chosen IMBs were deployed on level ice. I agree that this is a correct approach, to consider level ice only. However, the publicly available deployment documents for the buoys T63, T65, T70, and I1 indicate that ridged ice was already in close vicinity during deployment.

**Authors' response:** We acknowledge that the deployment sites of buoys T63, T65, T70, and I1 were close to ridged ice. However, our analysis shows that the increase in sea ice thickness was consistent and smooth, indicative of thermodynamic growth typical of level ice, as detailed in Fig. 4a. This suggests that the nearby ridged ice did not significantly impact the thermodynamic-growth characteristics for the level ice with the buoy deployments. We will increase our discussion to address these questions, mainly by comparing and analyzing the accumulation of snow at the buoy deployments sits and the growth rate of sea ice.

How was it ensured that ICESat-2 data was considered over level ice only? How long are the data segments, did they include only level data? While the modal value of the log-normal fitted freeboard is an estimate of the thermodynamically grown sea ice, it does not strictly exclude e.g. thin sea ice that has deformed and gained the same freeboard as thermodynamically grown undeformed sea ice.

**Authors' response:** We extracted level ice freeboards from over 15,000 IS2 freeboard segments within the MOSAiC DN scale, which includes a broad range of ice conditions. Our methodology did not exclusively filter for level ice; rather, it involved analyzing the entire freeboard distribution to isolate the level ice component, i.e. to determine the modal feature of the freeboard frequency distribution. While we recognize that deformed thin ice can influence the data, the statistical analysis at the 50 km scale ensures that such variations do not substantially impact the representation of level ice freeboards.

How about snow pits, have you considered that pressure ridge sites were sampled on MOSAiC, too? Level ice tends to have thinner snowpack with larger temperature gradients that lead to snow metamorphism affecting the snow density profile.

**Authors' response:** All snow pit measurements in this study was conducted on level ice, and we will clarify this in the revision.

When comparing your data to the AWI IceBird dataset, did you choose measurements on level ice only quality flag? I would suggest doing so, and in that case also using the updated version of the dataset.

**Authors' response:** We agree with your suggestion and will update the dataset in the revised manuscript, ensuring that only the level ice component is used for comparisons.

More specific comments:

L62ff: Alexandrov et al. (2010) did not use airborne multi-sensor data. They used ground-based drill-hole measurements achieved through landing airplanes on the sea ice in the 1980s (Soviet Sever expeditions). So far, I am not aware of any other study utilizing airborne multi-sensor measurements to derive sea-ice bulk density than Jutila et al. (2022).

**Authors' response:** Thanks to your clarification, we will rewrite the sentence.

L79ff: While Shi et al. (2023) have more recently argued the point, it was mentioned earlier in Jutila et al. (2022), to which also Shi et al. (2023) refer.

**Authors' response:** We apologize for the oversight regarding the details in your paper, and we will rewrite the sentence to include the insights of both Jutila et al. (2022) and Shi et al. (2023) in A10.

L179ff: Sea-ice freeboard and thickness are found to follow log-normal or exponential distribution, but does total freeboard behave the same? And how about on the 100 km scale?

**Authors' response:** First, the spatial scale of our study is 50 km. We will describe in detail the substantial evidence supporting the log-normal distribution of sea ice freeboard and thickness over tens of kilometers, including both in situ and airborne measurements.

L232ff: Both your snow depth and sea-ice thickness measurements come from the IMBs. Therefore, are their uncertainties not independent and the assumption thus wrong?

**Authors' response:** Although both sea ice thickness and snow depth data are sourced from IMBs, due to the independence of their respective observations, we consider the resulting uncertainties to be independent.

L244ff: Were any other formulations than first and second order polynomials investigated?

**Authors' response:** We explored both negative exponential and second order polynomial models to describe the relationship between univariate parameters and sea ice bulk density. As we found similar performance in both, we decided to use first and second order polynomials for formal consistency between univariate and bivariate parameterizations.

Figure 5 & L313ff: Which values are you using for the three J22 data points? To my eyes, they do not match the values from Table 3 in Jutila et al. (2022) that list the average bulk densities on 800 m spatial scale. Or did you perhaps derive those values from the nominal resolution datasets? Did you use all values or only the level ice ones? Furthermore, I recommend using the same marker shape for the same ice type, adding citations also to the main text, and explaining the acronym "J22" (now only on L361).

**Authors' response:** We used the mean values provided in Table 3 of Jutila et al. (2022). We agree with your suggestions and will make the modifications.

L480ff: The data consists of several profiles covering a total distance of more than 3000 km (3410 km). Surveyed sea ice was primarily first-year ice (100 % in 2017) and multi-year ice, with only very little second-year ice. While you mention the spatial resolution of the data, I also think it's important to distinguish between the nominal measurement spacing (5-6 m) and the

footprint size (40 m) of the measurement.

**Authors' response:** Thanks for the clarification, we will add this description in the revised version.

L493ff & Figure 10: Jutila et al. (2022) applied inverse-uncertainty weighted mean, not inverse distance. Are all ice types included in this analysis, also level ice and first-year ice, even though you're targeting to analyze rough and older ice? The AWI IceBird airborne sea-ice parameter datasets can easily distinguish different ice types using the provided quality flags.

**Authors' response:** We will correct the method description; this was a typo for which we apologise. In the revised manuscript, we will perform a more detailed analysis of ice types using the new version of the AWI IceBird dataset.

L523ff: The "new approach proposed in this study to determine [ice bulk density] at the basin scale using satellite altimetry data" is not new as this capability has been previously demonstrated in Jutila et al. (2022). If you mean using satellite altimetry data in your approach to determine sea-ice bulk density (together with ground-based point measurements), you need to present and discuss the effect of different scales for the reasons brought up earlier. The study also seems to highlight the parameterization applying the ice draft-to-thickness ratio, but there is no current or planned satellite mission that can directly observe sea-ice draft, thickness, nor their ratio.

**Authors' response:** We agree with your suggestion and will incorporate a discussion on how the differences in spatial scales affect sea ice bulk density retrieval. While no satellite missions currently or planned can directly observe sea ice draft and thickness, we will explore coupling the draught-to-thickness ratio with the hydrostatic equilibrium equation. This method will enable the simultaneous estimation of sea ice thickness and bulk density using satellite-measured freeboard and supplementary snow load data.

**Reference.**

[revised manuscript text omitted]

von Albedyll, L., Hendricks, S., Grodofzig, R., Krumpen, T., Arndt, S., Belter, H. J., Birnbaum, G., Cheng, B., Hoppmann, M., and Hutchings, J.: Thermodynamic and dynamic contributions to seasonal Arctic sea ice thickness distributions from airborne observations, Elem Sci Anth, 10, 00074, 2022.

---

## Author Comment (AC2)

**Responses to CC2 (Arttu Jutila)**
* * *
**Dear Arttu Jutila,**

Thank you for your valuable suggestions and the further discussions needed. We apologize for any previous misunderstandings concerning your remarks. We have made several significant revisions, as detailed in the attached document. The original comments are in black, and our replies are written in blue.

**Best regards,**
**Yi Zhou and other co-authors.**
* * *
Reply to author comment https://doi.org/10.5194/egusphere-2024-1240-AC1 by Zhou et al. for the manuscript by Zhou, Y., Wang, X., Lei, R., von Albedyll, L., Perovich, D. K., Zhang, Y., and Haas, C.: Seasonal evolution and parameterization of Arctic sea ice bulk density: results from the MOSAiC expedition and ICESat-2/ATLAS, EGUsphere [preprint], https://doi.org/10.5194/egusphere-2024-1240, 2024.

Dear Zhou et al.,

Thank you for your response to my comment and in particular for continuing the public discussion in the spirit of The Cryosphere's interactive review process.

I am pleased to see that you have carefully considered my comments. Nevertheless, I still see some loose ends which I would like to discuss further.

First, we would like to inform that we have significantly revised and re-evaluated some results, and further details will be provided in the revised manuscript. Given that you are a major contributor to the MOSAiC airborne data and the AWI Icebird multi-sensor sea ice dataset, we respectfully invite you to participate as a co-author on the revised manuscript, subject to policy permissions. Your involvement would greatly enhance the manuscript's quality. All modifications can be found in *Supplementary A1* at the end.

**Scales.** Thank you for the very helpful additional Figure A1. While you clarify that the extraction radii for IS2 and IMBs were 50 km and 30-40 km, respectively, you have unfortunately misread my comment. I did not use the term **radius** but **diameter**, i.e. two times the radius, for a very specific reason. Let's consider a rather standard satellite data product that has data in a regular grid format where the grid cells are square-shaped with each side measuring 25 km in length. This results in a spatial scale of 25 km – not 12.5 km that would be the radius of the largest possible circle drawn within the grid cell. In your case, for example looking at Figure 3c where the IS2 ground track passes nearly directly above the center of the circle or the MOSAiC CO, you end up extracting along-track data for a length of 100 km, not 50 km. That is the largest length scale of your input data. The same principle applies to the 40-m footprint size of the AWI IceBird measurements: it is the diameter, not the radius, of the EM-Bird footprint, within which snow depth and freeboard measurements are averaged and sea-ice bulk density is eventually derived.

In your consideration of the helicopter-borne laser scanner data in Hutter et al. (2023), you state having found it temporally insufficient with only about ten days of valid data for your study period. Did you include also the approximately weekly transect flights that extend beyond the MOSAiC CO floe for a few tens of kilometers into the DN, in most cases reaching the three L-sites? They cover several buoys, including seven IMBs of your study.

**Authors' response:** We apologize for the misunderstanding of your comments. Considering your suggestion, we included ALS airborne data and extended the analysis to the L-site scale **(as shown in Fig. A1)**. The DN scale was defined as a spatial scale with a radius of 50 km from the CO, while the L-site scale was defined as a radius of 25 km. We then used all IMBs within the L-site scale and the corresponding IS2/ALS modal freeboards to derive the IBD **(Fig. A2)**, with more details to be provided in the revised manuscript.

Furthermore, I would like to pose you a question: which of the two has more severe effects, lack of spatial or temporal overlap? Earlier you stated that during the winter season, which is the focus of your study, there was little sea-ice dynamics influencing the study area. Temporal interpolation should not cause too many problems then. After all, you already apply it for snow density. Figure 4d even shows that the IS2 modal freeboard values are very close to the trend line.

**Authors' response:** We suggest that the lack of spatial overlap is more severe. Due to the strong spatial heterogeneity of sea ice, the averaged results from the buoy arrays and the measurements at the IS2/ALS scales inevitably introduce spatial scale differences. This spatial variability poses challenges for the coordination of using both the average values from the IMB array and the modal values from IS2/ALS in the hydrostatic equilibrium equation. Therefore, in the revised manuscript, we introduced a spatial scale correction term to mitigate this issue. **Please refer to our response to RC1 for details.**

[Figure]

**Figure A1.** Distribution of IS2, ALS, and IMB array measurements, showing the cases on January 7, 2020, January 23, 2020, February 17, 2020, and April 23, 2020.

[Figure]

**Figure A2.** Comparison of L-site and DN sea ice parameters, including (a) sea ice thickness, (b) snow depth, (c) modal freeboard, and (d) sea ice bulk density. Statistical metrics include correlation coefficient (CC), mean difference (MD, L-site minus DN), and root mean square difference (RMSD).

**IS2 modal freeboard.** I may have been too quick previously agreeing that modal freeboard is an estimate of level ice. Coming back to it now after a while, I would like you to carefully consider the log-normal distribution and the modal value with respect to different variables. For sea-ice thickness measurements, it is indeed well-known and supported by studies that the modal value represents the most common value and the thickness of thermodynamically-grown level ice. For freeboard variables, whether it is sea-ice freeboard or snow/total freeboard, I think it is not so straight forward. In your answer to my comment, you give a long list of references very much like Koo et al. (2021), who write:

*"Since the modal thickness represents the thickness of the most frequently observed ice or level ice (Farrell et al., 2012; Hansen et al., 2013; Petty et al., 2016; Rack et al., 2021; Tian et al., 2020), we estimate the thermodynamic ice growth around the buoys by using the variations in the modal thickness."*

They, like many if not all the studies you referenced, first transform snow or sea-ice freeboard (depending on the sensor in question) into sea-ice thickness **with auxiliary data** using the hydrostatic equilibrium assumption before deriving the thickness distribution and its modal value. I suppose that looking only at freeboard distribution you are (1) mixing the terms snow freeboard and sea-ice freeboard and (2) "cutting corners" in your study. Are sea-ice and snow freeboards correlated with each other or with sea-ice thickness, do they always behave the same? Could a seemingly level snow freeboard conceal sea ice that is not level? A single snow freeboard value can correspond to a wide range of sea-ice thickness values depending on the snow load that cannot be deduced from laser altimetry alone. Let's consider a fixed total freeboard value of 0.3 m, approximately the derived modal value in your Figure 3c. Additionally, let's assign representative and fixed values for the densities of sea water, ice, and snow, but let snow depth vary from 0 to 0.3 m (it could be even larger for cases of negative freeboard). The sea-ice thickness values resulting from this single total freeboard value, but varying snow depth information, can vary by more than 1.5 m according to the hydrostatic equilibrium equation.

**Authors' response:** We fully understand your concern that the modal value of the total freeboard distribution cannot represent the average state of sea ice, as snow cover may affect the statistical properties of the sea ice fraction. However, we have not conflated the difference between total freeboard (which includes snow cover) and sea ice freeboard, and we expect that the impact of snow cover on the statistical characteristics of sea ice over scales

of tens of kilometers is very limited. To demonstrate the validity of the modal value of total freeboard distribution representing the average state of total freeboard for level ice, we used the latest Icebird total freeboard data with ice surface classification labels to evaluate our method. Based on the AWI Icebird measurements from April 2017 and April 2019, we compared the modal value of the total freeboard distribution (all ice types) with the mean total freeboard of level ice **(Fig. A3)**, retaining only the results for each measurement date where the proportion of level ice exceeds 20%. Overall, the two freeboard values are very close, with all proximity values greater than 90%, demonstrating the validity of our method.

[Figure]

**Figure A3.** Comparison of the modal value of total freeboard distribution (all ice types) with the mean total freeboard of level ice from AWI Icebird measurements in 2017 and 2019. The example of 2 April 2017 shows (a) the total freeboard distribution (b) the measurement area, and (c) the total freeboard profile. (d) Results for each measurement date, including the modal freeboard, mean value for level ice, absolute relative percentage difference between the two values (proximity), and the proportion of level ice (percentage).

**Snow pits.** Regarding the snow pit data, I do recommend being more explicit and more detailed in the methods section. While the data descriptor paper by Macfarlane et al. (2023) states a total of 576 snow pits, you have used only a small part of that data in your analysis. In fact, from the MOSAiC snow pit snow density cutter dataset (https://doi.org/10.1594/PANGAEA.940214), I can count only 85 snow pits between 25 October 2019 and 30 April 2020. The large number of "snow pits" stems from the fact how snow pits of different complexity were defined during the expedition: even a single profile with the SnowMicroPen (SMP) instrument could count as a snow pit. In the context of your manuscript, however, this is misleading since you did not use the SMP density profiles in your analysis. Furthermore, I trust you have indeed dug deep into the MOSAiC jargon of locations: snow pit locations "FR" in January-February and "DR" in April refer to "Fort Ridge" and "David's Ridge" sites, respectively, that do not represent level ice conditions.

**Authors' response:** Thank you for your clarification. We will provide a detailed description of the specific snow pits used in the revised manuscript. We have reviewed the snow pit results that included ice ridges (e.g. FR and DR) and excluded these records to recalculate the snow bulk density, see panel (c) of **Figure A4.**

[Figure]

**Figure A4.** Variations of sea ice and snow during the MOSAiC freezing season (DN scale). (a) Sea ice thickness measurements from the IMBs. The peculiarities of T72, I2, and I3 sites (as described in the text) are indicated by black arrows. The gray shaded band indicates the mean ± standard deviation (at least 10 buoys). (b) Snow depth measurements obtained from buoys and transects. The blue arrows mark four storm events, and the yellow arrow indicates a period characterized by a strong snow drifting event, according to Wagner et al. (2022). (c) Snow bulk density derived from snow pit measurements. (d) Seasonal evolution of the IS2 modal freeboard, with uncertainties indicated by error bands.

**AWI IceBird average densities.** In your response you claim to have used the mean values of sea-ice bulk density from Table 3 of Jutila et al. (2022). I have now double-checked this, and while the values for SYI and MYI seem fine, I don't think this is true for FYI. Table 3 of Jutila et al. (2022) states 929.3 +/- 16.0 kg m-3 for 2017 and 925.4 +/- 17.7 kg m-3 for 2019, in addition the main text states 928.5 +/- 16.4 kg m-3 as the overall average density for FYI. However, your code snippet "Sea_ice_bulk_density.m" in Zenodo (https://doi.org/10.5281/zenodo.11055727) shows that a value of 921.4222 +/- 18.5586 kg m-3, smaller than any other FYI average value given in Jutila et al. (2022), was used for plotting Figure 5.

**Authors' response:** Thank you for your clarification, and we apologize for the inaccuracy values that we used. In the revised manuscript, we calculated the mean IBD of SYI and FYI, including only the level ice portions, based on the latest version of the AWI Icebird data (original resolution). The mean and standard deviation of the recalculated IBD were ~913 kg m$^{-3}$ and 40 kg m$^{-3}$, respectively, and were compared with our results **(Fig. A5)**. Overall, the average IBD during the relatively stable phase of MOSAiC is very consistent with the average IBD from Icebird.

[Figure]

**Figure A5.** Seasonal evolution of IBD during the MOSAiC freezing season. The orange line indicates a significant decreasing trend in the MOSAiC IBD, while the green line indicates the mean value during a relatively stable phase. Also shown are the mean IBDs of FYI (black diamond) and MYI (black circle) from the A10 climatology (Alexandrov et al., 2010), the mean IBDs of FYI (cyan square) and MYI (diamond) estimated during the Arctic Sever expedition from 1980 to 1989 (Shi et al., 2023), the mean ice densities (FYI and MYI, purple triangle) from 2000 to 2015 based on in situ observations (Ji et al., 2021), and the mean IBD of FYI and SYI (diamond) based on AWI Icebird multi-sensor measurements in April 2017 and 2019 (Jutila et al., 2022). The underline indicates IBD results for level ice only. The gray error bars indicate the uncertainty of the MOSAiC IBD, and the other error bars indicate one standard deviation. Note that the density data from the historical measurements correspond to the month, regardless of the year.

**Reference.**

Alexandrov, V., Sandven, S., Wahlin, J., and Johannessen, O.: The relation between sea ice thickness and freeboard in the Arctic, The Cryosphere, 4, 373-380, 2010.

Ji, Q., Li, B., Pang, X., Zhao, X., and Lei, R.: Arctic sea ice density observation and its impact on sea ice thickness retrieval from CryoSat–2, Cold Regions Science and Technology, 181, 103177, 2021.

Jutila, A., Hendricks, S., Ricker, R., von Albedyll, L., Krumpen, T., and Haas, C.: Retrieval and parameterisation of sea-ice bulk density from airborne multi-sensor measurements, The Cryosphere, 16, 259-275, 2022.

Shi, H., Lee, S.-M., Sohn, B.-J., Gasiewski, A. J., Meier, W. N., Dybkjær, G., and Kim, S.-W.: Estimation of snow depth, sea ice thickness and bulk density, and ice freeboard in the Arctic winter by combining CryoSat-2, AVHRR, and AMSR measurements, IEEE Transactions on Geoscience and Remote Sensing, 2023. 2023.

Wagner, D. N., Shupe, M. D., Cox, C., Persson, O. G., Uttal, T., Frey, M. M., Kirchgaessner, A., Schneebeli, M., Jaggi, M., and Macfarlane, A. R.: Snowfall and snow accumulation during the MOSAiC winter and spring seasons, The Cryosphere, 16, 2022.

**[Data]**

**a) IS2 Freeboard Data Updated to ATL10 version 6.**

*Kwok et al. (2023), ATLAS/ICESat-2 L3A Sea Ice Freeboard, Version 6. [Data Set]. NSIDC.(https://doi.org/10.5067/ATLAS/ATL10.006).*

**b) Airborne Laser Scanning (ALS) Data: Added L-site scale data during the MOSAiC freezing season.**

*Hutter et al. (2023), Gridded segments from helicopter-borne laser scanner during MOSAiC. [PANGAEA](https://doi.org/10.1594/PANGAEA.950339).*

**c) AWI IceBird Multi-Sensor Sea Ice Parameters Updated to the Latest Version**

*- Jutila et al. (2024), Airborne sea ice parameters during the IceBird Winter 2019 campaign in the Arctic Ocean, Version 2. [PANGAEA](https://doi.org/10.1594/PANGAEA.966057).*
*- Jutila et al. (2024),Airborne sea ice parameters during the PAMARCMIP2017 campaign in the Arctic Ocean, Version 2. [PANGAEA](https://doi.org/10.1594/PANGAEA.966009).*

**d) Core-Based Ice Density Data: Added data on sea ice density from the *Sea Ice Physics Group* during MOSAiC legs 1 to 4.**

*- Oggier et al. (2023), First-year sea-ice salinity, temperature, density, oxygen and hydrogen isotope composition from the main coring site (MCS-FYI) during MOSAiC legs 1 to 4 in 2019/2020. [PANGAEA](https://doi.org/10.1594/PANGAEA.956732).*
*- Oggier et al. (2023), Second-year sea-ice salinity, temperature, density, oxygen and hydrogen isotope composition from the main coring site (MCS-SYI) during MOSAiC legs 1 to 4 in 2019/2020. [PANGAEA](https://doi.org/10.1594/PANGAEA.959830).*

**[Method]**

**a) IS2 Modal Freeboard Calculation:** In the revised manuscript, we have retained the original resolution of IS2 ATL10 v6 and no longer perform the 150-segment averaging. Additionally, we no longer use log-normal fitting to estimate modal freeboard. Instead, we directly use the average of the freeboard values corresponding to the five highest frequencies in the freeboard distribution to obtain the modal freeboard (standard deviation is used as the uncertainty for the quasi-peak region of the total freeboard distribution). The purpose of this modification is to preserve the original distribution characteristics of the data as much as possible, without imposing fitting constraints or making resolution adjustments. Moreover, we have adopted the same approach to obtain the modal ALS freeboard.

**b) Modal Value Feasibility for Level Ice:** The AWI Icebird dataset was utilized to assess the feasibility of our approach, which features rigorously defined ice surface classification labels. We obtained the modal freeboard from the total freeboard distribution that includes both level and rough ice, and compared it with the average total freeboard of level ice.

**c) Spatial Scale Correction:** We introduced a spatial scale correction term to better align buoy array data with IS2/ALS modal freeboards.

**d) Added Uncertainty for IS2 Modal Freeboard:** We added uncertainty to the IS2 modal freeboard by calculating the mean difference (~0.0125 m) between IS2 and reference ALS modal freeboard.

**[Results]**

**a) Sea Ice Bulk Density (IBD) Recalculation:** All IBD results have been recalculated and re-evaluated following substantial revisions.

**b) Enhanced IBD Parameterizations:** Additional details have been included in the IBD parameterizations to improve clarity and accuracy.

**c) IBD Results at Different Scales:** Besides the DN scale, IBD results have now been extended to include the L-site scale.

**d) High-Precision Ice Core Density:** Ice core density that we used were obtained using the high-precision hydrostatic weighing method.

**e) Expanded Discussion on IBD Uncertainty:** More comprehensive discussions have been added regarding the uncertainty of IBD, its seasonal variations, spatial heterogeneity, limitations, and potential applications.

---

## Author Comment (AC3)

**Responses to CC3 (Hoyeon Shi)**
* * *
**Dear Hoyeon Shi,**

Thank you for your interest in our work and for providing valuable suggestions. We have provided clarification and discussion regarding your questions, and we have included several significant revisions in the revised manuscript, as detailed in the attached document. The original comments are in black, and our replies are written in blue.

**Best regards,**
**Yi Zhou and other co-authors.**
* * *
Dear Zhou et al.,

Thank you for sharing the interesting results with the Cryosphere community. I enjoyed reading the manuscript, but I have two concerns, so I'm leaving them here. I would appreciate it if the authors could consider the following comments.

Firstly, we need to inform that we have made some major revisions, and all the modifications can be found in **Supplementary A1** at the end.

The first is about using the modal IS2 freeboard, which has already been raised in detail by "Reply on AC1" by Arttu Jutila. Although the authors provide two arguments in L179, the question that must be answered regarding this issue is "How could the modal freeboard be more physically compatible with the hydrostatic balance equation (used for IBD estimation) than the mean freeboard?" I would like to emphasize that my question is not related to the authors' assumption of the log-normal distribution of freeboard. This is because both "Mean" and "Mode" are available for any kind of statistical surface height distribution. What would be the physical meaning of modal height? How much do the parameterizations obtained from the mean and modal freeboards differ? The hydrostatic balance equation describes the balance of snow and ice mass and buoyancy, which are the quantities proportional to physical volume. Considering that the volume is area times height, it sounds more natural to me to multiply mean height than the modal height to area. I would like to hear the authors' opinions on this.

**Authors' response:** We will add more details in the **Methods** to further clarify your question. In this study, our primary aim is to match the IMB array and IS2 measurements to retrieve IBD. All selected buoys were deployed on level ice and showed good agreement with broader transect measurements, and thus we are confident that the buoy deployment strategy is sufficient to represent the level ice at the DN scale. On this basis, the mean ice thickness and snow depth from the buoy array (at least 10 buoys per day) were considered as the mean ice thickness and snow depth of level ice at the DN scale. However, for IS2 (or ALS) measurements, we cannot directly distinguish the total freeboard of level ice, thus we need to rely on the so-called "modal freeboard". By analyzing the frequency distribution of IS2 total freeboard within the DN region, we considered the quasi-frequency peak corresponding to the total freeboard as a proxy of the average total freeboard of level ice. We have validated the feasibility of this approach using AWI Icebird total freeboard data, which has a rigorous ice surface classification **(see our response to CC2 for details)**. Therefore, we actually matched the level ice thickness and snow depth from the buoy array with the total freeboard of level ice from IS2 modal characteristics. This, combined with auxiliary information on snow bulk density obtained from level ice, was used to calculate IBD based on the hydrostatic equilibrium equation.

In addition, a spatial scale correction term has been added to mitigate spatial scale differences between the buoy array and IS2 measurements **(see our response to RC1 for details)**.

Second, the manuscript is missing some context about using bivariate parameters for the IBD parameterization. The authors wrote they examined bivariate parameters, unlike previous studies that focused solely on the univariate parameters. This might mislead readers into thinking that there has been no study that used bivariate parameters to parameterize IBD, which is not true. The two papers already cited in the manuscript, Alexandrov et al. (2010) and Shi et al. (2023), have used the ice freeboard-to-thickness ratio to parameterize the sea ice bulk density. It should be noted that the 'ice draft-to-thickness ratio' is actually exactly the same as 1 – ice freeboard-to-thickness ratio. Furthermore, since Eq. (8) of the manuscript is a first-order linear equation, the form of the parameterization suggested by this study is the same as the one previously suggested by the two papers, i.e.:

IBD = a1 * draft-to-thickness-ratio + a2 = a1 * (1 – freeboard-to-thickness-ratio) + a2 = -a1 * freeboard-to-thickness-ratio + (a1 + a2)

Previous studies interpreted "-a1" as the density difference between floating and submerged parts of sea ice and "a1 + a2" as the density of submerged part of sea ice. Moreover, this parameterization has been implemented in the simultaneous estimation method that consistently estimates snow depth, sea ice thickness, ice freeboard, and IBD (Shi et al., 2023). Although the authors wrote in L358, "The strong linear relationship between the two bivariate parameters and IBD …, supporting previous suggestions (Alexandrov et al., 2010; Shi et al., 2023)", I would recommend authors to provide a more complete context of using bivariate parameters in section 2.2.4.

Nevertheless, I think this study's novelty comes from determining coefficients of parameterization based on the multisource observations rather than using representative values available from the literature. Accordingly, it would be very valuable to our community if the authors could include the following things.

(1) Parameterization of IBD using the ice freeboard-to-thickness ratio, i.e.:

IBD = a1 * freeboard-to-thickness-ratio + a2

(2) Comparison of the determined parameterization equation above and the equations used in the previous studies with physical interpretation of the difference between them.

For instance, in Alexandrov et al. (2010), for multiyear sea ice, a1 is -370 kg m$^{-3}$ (= 550 – 920) and a2 is 920 kg m$^{-3}$. In Shi et al. (2023), a2 is the same, while a1 is -105 kg m-3 (= 815 – 920) for multiyear sea ice and -45 kg m$^{-3}$ (= 875 – 920) for first-year sea ice.

**Authors' response:** Thank you for your clarification, and we will include background information on using bivariate parameters for IBD parameterization in the revised manuscript. So far, we have added more detail to the IBD parameterization, and beyond the ice draft-thickness ratio (which shows a strong linear relationship with IBD), we have considered three fitting relationships for the other parameters: quadratic polynomial, exponential, and power equation **(Fig. A1)**. Considering that the draft-to-thickness ratio can be formally converted to the freeboard-to-thickness ratio, we also provided the coefficients for the latter in the revised manuscript. Furthermore, following your suggestion, we compared the new IBD parameterizations with previous studies **(Fig. A2)**. The scope of the comparison was constrained within the valid range of input parameters used in this study. More details regarding the comparisons will be provided in the revised manuscript; however, it is crucial to clarify that the

regression coefficients of the ice freeboard-to-thickness parameterization we derived are solely statistical significance, differing fundamentally from the coefficients obtained by A10 and S23 based on the reference IBDs used for weighting.

[Figure]

**Figure A1.** Parameterization of the IBD, including regression models using (a) sea ice thickness, (b) total freeboard, (c) sea ice draft, (d) sea ice freeboard, (e) ice freeboard-to-total freeboard ratio, and (f) ice draft-to-thickness ratio. Each panel shows model fit metrics, including the coefficient of determination ($R^2$) and the root mean square error ($RMSE$). Note that the statistical $P$-value for all results is less than 0.05 and the unit of $RMSE$ is in kg m$^{-3}$.

[Figure]

**Figure A2.** Intercomparison of sea ice bulk density parameterizations. (a) Sea ice thickness-based parameterizations, where the black line represents the results from Kovacs (1997) (K97) and the other colored lines are from this study. (b) Sea ice freeboard-based parameterizations, where the black line represents the results from Jutila et al. (2022) (J22) and the other colored lines are from this study. (c) The parameterizations based on ice freeboard-to-thickness ratio, where the black line represents the MYI results from Alexandrov et al. (2010) (A10), green and blue lines represent the FYI and MYI results from Shi et al. (2023) (S23) ,respectively, and red line is from this study.

**Reference.**

Alexandrov, V., Sandven, S., Wahlin, J., and Johannessen, O.: The relation between sea ice thickness and freeboard in the Arctic, The Cryosphere, 4, 373-380, 2010.

Jutila, A., Hendricks, S., Ricker, R., von Albedyll, L., Krumpen, T., and Haas, C.: Retrieval and parameterisation of sea-ice bulk density from airborne multi-sensor measurements, The Cryosphere, 16, 259-275, 2022.

Kovacs, A.: Estimating the full-scale flexural and compressive strength of first-year sea ice, Journal of Geophysical Research: Oceans, 102, 8681-8689, 1997.

Shi, H., Lee, S.-M., Sohn, B.-J., Gasiewski, A. J., Meier, W. N., Dybkjær, G., and Kim, S.-W.: Estimation of snow depth, sea ice thickness and bulk density, and ice freeboard in the Arctic winter by combining CryoSat-2, AVHRR, and AMSR measurements, IEEE Transactions on Geoscience and Remote Sensing, 2023. 2023.

**a) IS2 Freeboard Data Updated to ATL10 version 6.**

*Kwok et al. (2023), ATLAS/ICESat-2 L3A Sea Ice Freeboard, Version 6. [Data Set]. NSIDC.(https://doi.org/10.5067/ATLAS/ATL10.006).*

**b) Airborne Laser Scanning (ALS) Data: Added L-site scale data during the MOSAiC freezing season.**

*Hutter et al. (2023), Gridded segments from helicopter-borne laser scanner during MOSAiC. [PANGAEA](https://doi.org/10.1594/PANGAEA.950339).*

**c) AWI IceBird Multi-Sensor Sea Ice Parameters Updated to the Latest Version**

*- Jutila et al. (2024), Airborne sea ice parameters during the IceBird Winter 2019 campaign in the Arctic Ocean, Version 2. [PANGAEA](https://doi.org/10.1594/PANGAEA.966057).*
*- Jutila et al. (2024),Airborne sea ice parameters during the PAMARCMIP2017 campaign in the Arctic Ocean, Version 2. [PANGAEA](https://doi.org/10.1594/PANGAEA.966009).*

**d) Core-Based Ice Density Data: Added data on sea ice density from the *Sea Ice Physics Group* during MOSAiC legs 1 to 4.**

*- Oggier et al. (2023), First-year sea-ice salinity, temperature, density, oxygen and hydrogen isotope composition from the main coring site (MCS-FYI) during MOSAiC legs 1 to 4 in 2019/2020. [PANGAEA](https://doi.org/10.1594/PANGAEA.956732).*
*- Oggier et al. (2023), Second-year sea-ice salinity, temperature, density, oxygen and hydrogen isotope composition from the main coring site (MCS-SYI) during MOSAiC legs 1 to 4 in 2019/2020. [PANGAEA](https://doi.org/10.1594/PANGAEA.959830).*

**a) IS2 Modal Freeboard Calculation:** In the revised manuscript, we have retained the original resolution of IS2 ATL10 v6 and no longer perform the 150-segment averaging. Additionally, we no longer use log-normal fitting to estimate modal freeboard. Instead, we directly use the average of the freeboard values corresponding to the five highest frequencies in the freeboard distribution to obtain the modal freeboard (standard deviation is used as the uncertainty for the quasi-peak region of the total freeboard distribution). The purpose of this modification is to preserve the original distribution characteristics of the data as much as possible, without imposing fitting constraints or making resolution adjustments. Moreover, we have adopted the same approach to obtain the modal ALS freeboard.

**b) Modal Value Feasibility for Level Ice:** The AWI Icebird dataset was utilized to assess the feasibility of our approach, which features rigorously defined ice surface classification labels. We obtained the modal freeboard from the total freeboard distribution that includes both level and rough ice, and compared it with the average total freeboard of level ice.

**c) Spatial Scale Correction:** We introduced a spatial scale correction term to better align buoy array data with IS2/ALS modal freeboards.

**d) Added Uncertainty for IS2 Modal Freeboard:** We added uncertainty to the IS2 modal freeboard by calculating the mean difference (~0.0125 m) between IS2 and reference ALS modal freeboard.

**[Results]**

**a) Sea Ice Bulk Density (IBD) Recalculation:** All IBD results have been recalculated and re-evaluated following substantial revisions.

**b) Enhanced IBD Parameterizations:** Additional details have been included in the IBD parameterizations to improve clarity and accuracy.

**c) IBD Results at Different Scales:** Besides the DN scale, IBD results have now been extended to include the L-site scale.

**d) High-Precision Ice Core Density:** Ice core density that we used were obtained using the high-precision hydrostatic weighing method.

**e) Expanded Discussion on IBD Uncertainty:** More comprehensive discussions have been added regarding the uncertainty of IBD, its seasonal variations, spatial heterogeneity, limitations, and potential applications.

---

## Author Comment (AC4)

**Responses to RC1**
* * *
**Dear reviewer,**

We sincerely appreciate your constructive comments, which have significantly contributed to the improvement of our manuscript. We have made thorough and detailed revisions according to your suggestions. Please refer to the attached document for a detailed review.

**Best regards,**
**Yi Zhou and other co-authors.**
* * *
This manuscript, titled "Seasonal evolution and parameterization of Arctic sea ice bulk density: results from the MOSAiC expedition and ICESat-2/ATLAS", deals with the sea ice density, which is a key issue in estimating the sea ice thickness in satellite altimetry. I consider it novel to use MOSAiC data for informing us of the seasonal evolution of sea ice density. And the majority of the analysis and result is sound. However, several major issues have to be made, which are listed below.

**Authors' response:** Thank you for your thorough analysis and insightful feedback on our manuscript. We have carefully considered and addressed each of the major issues you highlighted. Additionally, we have integrated suggestions from other members of the *EGU Community* to further enhance the quality and accuracy of our work. Overall, we have reassessed all the results and a brief list of refinements can be found in ***Supplementary A1*** (at the end).

First, the spatial representation issue is central to the analysis, and has to be dealt with in a more systematic way. The two particular sources of uncertainty in Eqs. 5 are that of hf (total freeboard) and rho_s (snow density). For example, for hf and the analysis with buoy-measured hi and hs in Fig. 3, the uncertainty is actually two fold, under formal definitions. First, the uncertainty between the mode of the log-normal fitted IS2 hf and the areal mean level-ice hf, and second, that between the areal mean level-ice hf and that measured at the buoy (in order to compare with buoy hi & hs). The total uncertainty addressed here is only the difference between fitted mode and the maximum probability bin of hf. This uncertainty falls into part of the first uncertainty I mentioned, and hence the second uncertainty is not accounted for. The representation error in snow depth is potentially large as well, but could diminish more quickly at larger scales. The representation issue is also raised by two community comments.

**Authors' response:** We agree with the reviewer's suggestion that the spatial representation of buoy arrays and satellite measurements needs to be better coordinated. In addition, we recognize that the uncertainties in the hf (total freeboard) used as an input to the hydrostatic equilibrium equation need further investigation. In the revised manuscript, we have made significant revisions in this regard, which can be summarized as follows:

**a) We evaluated the feasibility of using the modal value of the total freeboard distribution (including all ice types) as a proxy for the mean total freeboard of the level ice.** Specifically, we used the latest AWI Icebrid total freeboard data provided by Jutila et al. (2024a) and Jutila et al. (2024b) for detailed analysis and found that the modal freeboard and the mean freeboard for level ice were in very good agreement, demonstrating the robustness of our methodology **(Fig. A1, more details in our responces to CC2)**. This is the first step of our IBD retrieval, where we extract the level ice fraction from the IS2 measurements (i.e., IS2 modal freeboard) to ensure that the measured

sea ice type is consistent with the observations from the IMB array, as all selected IMBs are deployed on the level ice.

[Figure]

**Figure A1.** Comparison of the modal value of the total freeboard distribution (all ice types) with the mean total freeboard of level ice from AWI Icebird measurements in 2017 and 2019. The example for 2 April 2017 shows (a) the total freeboard distribution, (b) the measurement area, and (c) the total freeboard profile. (d) Results for each measurement date, including modal freeboard, mean level ice, absolute relative percentage difference between the two (proximity), and proportion of level ice (percentage).

**b) We updated the calculation method for the modal freeboard, thereby incorporating a portion of the Type I uncertainty of hf due to the quasi-peak frequency region.** In the revised manuscript, we have updated the IS2 data to version 6 and have discontinued the use of 150-segment averaging and log-normal fitting to determine IS2 modal values. Instead, we have maintained the original resolution of the IS2 data and determined the final IS2 modal freeboard by averaging the top five frequency peaks, with the standard deviation representing the Type I uncertainty. This modification aims to avoid introducing additional fitting and averaging constraints that could impact the original distribution characteristics of the IS2 data. Figure A2 presents several examples of the IS2 freeboard distribution, illustrating the modal freeboard (red line) and its associated uncertainty(red shaded band). In addition, we also introduced airborne laser scanning (ALS) data during MOSAiC to complement the L-site scale (25km radius from CO) analysis (see also Fig. A2).

[Figure]

**Figure A2.** Distribution of IS2, ALS, and IMB array measurements, showing the cases on January 7, 2020, January 23, 2020, February 17, 2020, and April 23, 2020.

**c) The difference between the IS2 modal freeboard and the reference modal freeboard was used as a supplement to the Type I uncertainty.** With the introduction of ALS data, which offered in-situ measurements with higher resolution and accuracy compared to IS2, we used the ALS modal freeboard as a reference to evaluate the IS2 modal freeboard. However, it must be acknowledged that airborne data also have inherent biases, but using airborne data to evaluate satellite data is a common practice. Considering that the ALS data only cover the L-site scale, we screened all IS2 data for dates with sufficient data points (more than 15,000 segments) for comparison. Ultimately, four effective date records were identified, as shown in Figure A2. We calculated the average deviation of the modal freeboard between ALS and IS2 for these dates, estimated to be ~0.0125 m, and incorporated this into the Type I uncertainty for IS2 modal freeboard.

**d) Coordination of spatial scales between the buoy array and IS2 measurements (Type II uncertainty).** To retrieve IBD based on the hydrostatic equilibrium equation, the satellite (or airborne) modal freeboard and the mean ice thickness and snow depth from the buoys need to achieve spatial coordination. However, to directly examine the spatial scale differences between the buoy array and satellite measurements, it is necessary to ensure that both scales have the same type of sea ice parameters (and at least some records), but this condition is currently not met. Therefore, we adopted an indirect method by introducing a spatial scale correction term to harmonize the

measurements from the two scales, and thus the type II uncertainty was considered as a correction term. The details are as follows:

**(1)** We compared the buoy array data with transects covering a larger measurement area and found that the trends in mean snow depth (surface ice only) were very close at both scales (Fig. A2(c)). Furthermore, according to Koo et al. (2021), the trend in the IS2-derived mode thickness was also found to be very close to that of the average thickness measured by the buoy array. Therefore, we expect that the buoy array will be able to capture variations in sea ice and snow over a spatial area larger than the buoy deployment sites, which serves as a fundamental basis for our subsequent spatial correction.

[Figure]

**Figure A3.** Variations of sea ice and snow during the MOSAiC freezing season (DN scale). (a) Sea ice thickness measurements from the IMBs. The peculiarities of T72, I2, and I3 sites (as described in the text) are indicated by black arrows. The gray shaded band indicates the mean ± standard deviation (at least 10 buoys). (b) Snow depth measurements obtained from buoys and transects. The blue arrows mark four storm events, and the yellow arrow indicates a period characterized by a strong snow drifting event, according to Wagner et al. (2022). (c) Snow bulk density derived from snow pit measurements. (d) Seasonal evolution of the IS2 modal freeboard, with uncertainties indicated by error bands.

(2) We subtracted the mean snow depth of the buoy array from the IS2 mode freeboard to evaluate the resulting sea ice freeboard. Subsequently, we observed a significant negative sea ice freeboard in early autumn, with a trend towards decreasing to zero before gradually increasing again. Based on the ice core data obtained from Oggier et al. (2023a) and Oggier et al. (2023b) (including SYI and FYI), we found that the mean sea ice freeboard was approximately 0.031 m in autumn. Therefore, this indicates that there are indeed systematic differences between the measurements of the buoy array and IS2. Considering their ability to capture similar trends in sea ice variations, we propose using the maximum negative sea ice freeboard and the baseline freeboard (derived from ice core averages) as an empirical correction term to account for the scale differences between the IS2 mode freeboard and the buoy array measurements (simply understood as a systematic uplift of the derived sea ice freeboard). Ultimately, when used for IBD retrieval, the IS2 modal freebord added an additional 0.045 m + 0.031 m to mitigate the issue of scale differences. For the ALS, there is a slight difference, represented as 0.046 m + 0.031 m. The recalculated DN scale IBD variations are shown in Fig. A4.

[Figure]

**Figure A4.** Seasonal evolution of IBD during the MOSAiC freezing season. The orange line indicates a significant decreasing trend in the MOSAiC IBD, while the green line indicates the mean value during a relatively stable phase. Also shown are the mean IBDs of FYI (black diamond) and MYI (black circle) from the A10 climatology (Alexandrov et al., 2010), the mean IBDs of FYI (cyan square) and MYI (diamond) estimated during the Arctic Sever expedition from 1980 to 1989 (Shi et al., 2023), the mean ice densities (FYI and MYI, purple triangle) from 2000 to 2015 based on in situ observations (Ji et al., 2021), and the mean IBD (FYI and SYI, yellow circle) based on AWI Icebird multi-sensor measurements in April 2017 and 2019 (Jutila et al., 2022). The underline indicates IBD results for level ice only. The gray error bars indicate the uncertainty of the MOSAiC IBD, and the other error bars indicate one standard deviation. Note that the density data from the historical measurements correspond to the month, regardless of the year.

Second, and consequently from the first point I raised, the apparent better fitting with the bivariate formulation (Sec. 3.3 and Fig. 7). One should be very careful in claiming that any fitting is better for parameterizing the ice density. Since if one look closely at Eqs. 5 and the first point I raised, it is immediate that the representation uncertainty will be a major source of correlation ($R^2 > 0.9$ for both cases) with bivariate formulation, since: rho_i = rho_w*(hi+hs-hf)/hi + ⋯, where (hi+hs-hf) is the derived sea ice draft, and contains a large representation error. This error gets carried to rho_i, in proportions, so that the values on both sides correlates really well ($R^2 = 0.9$). In a sense rho_i and draft/thickness ratio are NOT independent due to the way rho_i is derived. And more importantly, the representation uncertainty dominates over the variability in the snow-related, second term in Eqs. 5. Let me be very clear here: I think there should exist significant correlation between the rho_i and draft/thickness ratio, which could arises from physical reasons (see also IceBird results in Sec. 4.4). I just don't consider the argument here to be strict enough for comparing the parameterization schemes for ice density, and especially, whether the bivariates are better. Better quantification of uncertainty due to limited representation, is potentially needed before such claims.

**Authors' response:** We agree with your point that the better fit of the bivariate parameters does not necessarily mean that they are the best parameters to determine IBD. We will thoroughly discuss issues related to fit uncertainty, physical dependence and error propagation in the revised manuscript. However, based on the significant revisions in the IS2 modal freeboard and associated uncertainties, we have updated the IBD parameterisation scheme and provided more details. The new IBD parameterisations show a generally reduced correlation compared to the

original results (Fig. A5). This implies that the added uncertainty and scale corrections have reduced the error dependence between the sea ice parameters and the retrieved IBD.

[Figure]

**Figure A5.** Parameterization of the IBD, including regression models using (a) sea ice thickness, (b) total freeboard, (c) sea ice draft, (d) sea ice freeboard, (e) ice freeboard-to-total freeboard ratio, and (f) ice draft-to-thickness ratio. Each panel shows model fit metrics, including the coefficient of determination ($R^2$) and the root mean square error ($RMSE$). Note that the statistical $P$-value for all results is less than 0.05 and the unit of $RMSE$ is in kg m$^{-3}$.

Third, I think better sea ice topography data collected during MOSAiC campaign serve as a very good source of information for this study, which is a point already raised by Arttu (in first CC). However, maybe the dataset does not fully support the study of the whole winter, but it is definitely worth to look into and discussed in the paper.

**Authors' response:** We agree with the reviewer's suggestions. In the revised manuscript, we have included the MOSAiC ALS data and additionally retrieved the IBD results from the L-site scale (Fig. A2). We used all available buoys within the L-site scale as well as the available IS2 and ALS modal freeboards to retrieve IBD. On this basis, we compared the different sea ice parameters from the DN scale and the L-site scale (Fig. A6).

[Figure]

**Figure A6.** Comparison of L-site and DN sea ice parameters, including (a) sea ice thickness, (b) snow depth, (c) modal freeboard, and (d) sea ice bulk density. Statistical metrics include correlation coefficient (CC), mean difference (MD, L-site minus DN), and root mean square difference (RMSD).

Fourth, I consider the use of IceBird data could be improved. The scale dependency analysis is nice, but one has to be clear of two aspects. First, the uncertainty of SnowRadar (hence hs) and EM (hence hi+hs) over rough ice, which could compromise the analysis for this particularly important ice type. Second, the potential of apparent but superficial statistical correlation since the derived rho_i carries the measurement and representation error of the original measurements. Therefore, I suggest to change the analysis to level ice only with IceBird data, since such type info is available.

**Authors' response:** Thank you for your comments and suggestions. We have revised our use of the AWI Icebird data accordingly. We have re-calculated the mean IBD of SYI and FYI, including only level ice, from the Icebird data to compare with our results, as shown in Fig. A4. Overall, the IBD from MOSAiC during the relatively stable phase is very close to the IBD from Icebird. Furthermore, in the further analysis of the IBD parameterization using the Icebird sea ice parameters, we have detailed the results for different sea ice types (FYI, SYI and MYI), surface characteristics (rough and level ice), and spatial scales (40 m, 800 m, 12.5 km and 25 km), as shown in Fig. A7 and Fig. A8.

A minor comment: on line 452: the increase of hf may well be due to thermodynamic growth of ice, but purely/largely due to snow accumulation. So be more strict, as follows: These findings indicate that the magnitude of sea ice elevation changes exhibits significant spatial variability, possibly related to initial ice thickness, sea ice growth, and snow accumulation.

**Authors' response:** Thank you for your clarification and suggestions. We have reorganized this sentence in the revised manuscript.

[Figure]

**Figure A7.** Fitting performance of the ice draft-to-thickness ratio to the IBD at different sea ice types and surface characteristics based on AWI Icebird multi-sensor sea ice data (original resolution), including the results from (a) deformed and all ice types, (b) level and all ice types, (c) deformed and FYI, (d) level and FYI, (e) deformed and SYI, (f) level and SYI, (g) deformed and MYI, and (h) level and MYI.

[Figure]

**Figure A8.** Fitting performance of the ice draft-to-thickness ratio to the IBD at different spatial scales based on AWI Icebird multi-sensor sea ice data, including the results from (a) 40 m, (b) 800 m, (c) 12.5 km, and (d) 25 km.

**a) IS2 Freeboard Data Updated to ATL10 version 6.**
*Kwok et al. (2023), ATLAS/ICESat-2 L3A Sea Ice Freeboard, Version 6. [Data Set]. NSIDC.(https://doi.org/10.5067/ATLAS/ATL10.006).*

**b) Airborne Laser Scanning (ALS) Data: Added L-site scale data during the MOSAiC freezing season.**
*Hutter et al. (2023), Gridded segments from helicopter-borne laser scanner during MOSAiC. [PANGAEA](https://doi.org/10.1594/PANGAEA.950339).*

**c) AWI IceBird Multi-Sensor Sea Ice Parameters Updated to the Latest Version**
*- Jutila et al. (2024), Airborne sea ice parameters during the IceBird Winter 2019 campaign in the Arctic Ocean, Version 2. [PANGAEA](https://doi.org/10.1594/PANGAEA.966057).*
*- Jutila et al. (2024),Airborne sea ice parameters during the PAMARCMIP2017 campaign in the Arctic Ocean, Version 2. [PANGAEA](https://doi.org/10.1594/PANGAEA.966009).*

**d) Core-Based Ice Density Data: Added data on sea ice density from the *Sea Ice Physics Group* during MOSAiC legs 1 to 4.**
*- Oggier et al. (2023), First-year sea-ice salinity, temperature, density, oxygen and hydrogen isotope composition from the main coring site (MCS-FYI) during MOSAiC legs 1 to 4 in 2019/2020. [PANGAEA](https://doi.org/10.1594/PANGAEA.956732).*
*- Oggier et al. (2023), Second-year sea-ice salinity, temperature, density, oxygen and hydrogen isotope composition from the main coring site (MCS-SYI) during MOSAiC legs 1 to 4 in 2019/2020. [PANGAEA](https://doi.org/10.1594/PANGAEA.959830).*

**a) IS2 Modal Freeboard Calculation:** In the revised manuscript, we have retained the original resolution of IS2 ATL10 v6 and no longer perform the 150-segment averaging. Additionally, we no longer use log-normal fitting to estimate modal freeboard. Instead, we directly use the average of the freeboard values corresponding to the five highest frequencies in the freeboard distribution to obtain the modal freeboard (standard deviation is used as the uncertainty for the quasi-peak region of the total freeboard distribution). The purpose of this modification is to preserve the original distribution characteristics of the data as much as possible, without imposing fitting constraints or making resolution adjustments. Moreover, we have adopted the same approach to obtain the modal ALS freeboard.

**b) Modal Value Feasibility for Level Ice:** The AWI Icebird dataset was utilized to assess the feasibility of our approach, which features rigorously defined ice surface classification labels. We obtained the modal freeboard from the total freeboard distribution that includes both level and rough ice, and compared it with the average total freeboard of level ice.

**c) Spatial Scale Correction:** We introduced a spatial scale correction term to better align buoy array data with IS2/ALS modal freeboards.

**d) Added Uncertainty for IS2 Modal Freeboard:** We added uncertainty to the IS2 modal freeboard by calculating the mean difference (~0.0125 m) between IS2 and reference ALS modal freeboard.

**[Results]**

**a) Sea Ice Bulk Density (IBD) Recalculation:** All IBD results have been recalculated and re-evaluated following substantial revisions.

**b) Enhanced IBD Parameterizations:** Additional details have been included in the IBD parameterizations to improve clarity and accuracy.

**c) IBD Results at Different Scales:** Besides the DN scale, IBD results have now been extended to include the L-site scale.

**d) High-Precision Ice Core Density:** Ice core density that we used were obtained using the high-precision hydrostatic weighing method.

**e) Expanded Discussion on IBD Uncertainty:** More comprehensive discussions have been added regarding the uncertainty of IBD, its seasonal variations, spatial heterogeneity, limitations, and potential applications.

**Reference.**

Alexandrov, V., Sandven, S., Wahlin, J., and Johannessen, O.: The relation between sea ice thickness and freeboard in the Arctic, The Cryosphere, 4, 373-380, 2010.

Ji, Q., Li, B., Pang, X., Zhao, X., and Lei, R.: Arctic sea ice density observation and its impact on sea ice thickness retrieval from CryoSat–2, Cold Regions Science and Technology, 181, 103177, 2021.

Jutila, A., Hendricks, S., Ricker, R., von Albedyll, L., and Haas, C.: Airborne sea ice parameters during the IceBird Winter 2019 campaign in the Arctic Ocean, Version 2. PANGAEA, 2024a.

Jutila, A., Hendricks, S., Ricker, R., von Albedyll, L., and Haas, C.: Airborne sea ice parameters during the PAMARCMIP2017 campaign in the Arctic Ocean, Version 2. PANGAEA, 2024b.

Jutila, A., Hendricks, S., Ricker, R., von Albedyll, L., Krumpen, T., and Haas, C.: Retrieval and parameterisation of sea-ice bulk density from airborne multi-sensor measurements, The Cryosphere, 16, 259-275, 2022.

Koo, Y., Lei, R., Cheng, Y., Cheng, B., Xie, H., Hoppmann, M., Kurtz, N. T., Ackley, S. F., and Mestas-Nuñez, A. M.: Estimation of thermodynamic and dynamic contributions to sea ice growth in the Central Arctic using ICESat-2 and MOSAiC SIMBA buoy data, Remote Sensing of Environment, 267, 112730, 2021.

Oggier, M., Salganik, E., Whitmore, L., Fong, A. A., Hoppe, C. J. M., Rember, R., Høyland, K. V., Divine, D. V., Gradinger, R., Fons, S. W., Abrahamsson, K., Aguilar-Islas, A. M., Angelopoulos, M., Arndt, S., Balmonte, J. P., Bozzato, D., Bowman, J. S., Castellani, G., Chamberlain, E., Creamean, J., D'Angelo, A., Damm, E., Dumitrascu, A., Eggers, S. L., Gardner, J., Grosfeld, L., Haapala, J., Immerz, A., Kolabutin, N., Lange, B. A., Lei, R., Marsay, C. M., Maus, S., Müller, O., Olsen, L. M., Nuibom, A., Ren, J., Rinke, A., Sheikin, I., Shimanchuk, E., Snoeijs-Leijonmalm, P., Spahic, S., Stefels, J., Torres-Valdés, S., Torstensson, A., Ulfsbo, A., Verdugo, J., Vortkamp, M., Wang, L., Webster, M., Wischnewski, L., and Granskog, M. A.: First-year sea-ice salinity, temperature, density, oxygen and hydrogen isotope composition from the main coring site (MCS-FYI)

during MOSAiC legs 1 to 4 in 2019/2020. PANGAEA, 2023a.

Oggier, M., Salganik, E., Whitmore, L., Fong, A. A., Hoppe, C. J. M., Rember, R., Høyland, K. V., Gradinger, R., Divine, D. V., Fons, S. W., Abrahamsson, K., Aguilar-Islas, A. M., Angelopoulos, M., Arndt, S., Balmonte, J. P., Bozzato, D., Bowman, J. S., Castellani, G., Chamberlain, E., Creamean, J., D'Angelo, A., Damm, E., Dumitrascu, A., Eggers, L., Gardner, J., Grosfeld, L., Haapala, J., Immerz, A., Kolabutin, N., Lange, B. A., Lei, R., Marsay, C. M., Maus, S., Olsen, L. M., Müller, O., Nuibom, A., Ren, J., Rinke, A., Sheikin, I., Shimanchuk, E., Snoeijs-Leijonmalm, P., Spahic, S., Stefels, J., Torres-Valdés, S., Torstensson, A., Ulfsbo, A., Verdugo, J., Vortkamp, M., Wang, L., Webster, M., Wischnewski, L., and Granskog, M. A.: Second-year sea-ice salinity, temperature, density, oxygen and hydrogen isotope composition from the main coring site (MCS-SYI) during MOSAiC legs 1 to 4 in 2019/2020. PANGAEA, 2023b.

Shi, H., Lee, S.-M., Sohn, B.-J., Gasiewski, A. J., Meier, W. N., Dybkjær, G., and Kim, S.-W.: Estimation of snow depth, sea ice thickness and bulk density, and ice freeboard in the Arctic winter by combining CryoSat-2, AVHRR, and AMSR measurements, IEEE Transactions on Geoscience and Remote Sensing, 2023. 2023.

Wagner, D. N., Shupe, M. D., Cox, C., Persson, O. G., Uttal, T., Frey, M. M., Kirchgaessner, A., Schneebeli, M., Jaggi, M., and Macfarlane, A. R.: Snowfall and snow accumulation during the MOSAiC winter and spring seasons, The Cryosphere, 16, 2022.